# Towards Reliable Detection of Empty Space: Conditional Marked Point Processes for Object Detection

**Tobias J. Riedlinger** *
Technical University of Berlin
riedlinger@math.tu-berlin.de

**Kira Maag** *
Heinrich-Heine-University Düsseldorf
kira.maag@hhu.de

**Hanno Gottschalk**
Technical University of Berlin
gottschalk@math.tu-berlin.de

## Abstract

Deep neural networks have set the state-of-the-art in computer vision tasks such as bounding box detection and semantic segmentation. Object detectors and segmentation models assign confidence scores to predictions, reflecting the model's uncertainty in object detection or pixel-wise classification. However, these confidence estimates are often miscalibrated, as their architectures and loss functions are tailored to task performance rather than probabilistic foundation. Even with well calibrated predictions, object detectors fail to quantify uncertainty outside detected bounding boxes, i.e., the model does not make a probability assessment of whether an area without detected objects is truly free of obstacles. This poses a safety risk in applications such as automated driving, where uncertainty in empty areas remains unexplored. In this work, we propose an object detection model grounded in spatial statistics. Bounding box data matches realizations of a marked point process, commonly used to describe the probabilistic occurrence of spatial point events identified as bounding box centers, where marks are used to describe the spatial extension of bounding boxes and classes. Our statistical framework enables a likelihood-based training and provides well-defined confidence estimates for whether a region is drivable, i.e., free of objects. We demonstrate the effectiveness of our method through calibration assessments and evaluation of performance.

## 1 Introduction

Deep neural networks (DNNs) have demonstrated outstanding performance in computer vision tasks like image classification (Wortsman et al., 2022), object detection (Zong et al., 2023), instance segmentation (Yan et al., 2023) and semantic segmentation (Xu et al., 2023). Object detection describes the task of identifying and localizing objects of a particular class, for example, by predicting bounding boxes or by labeling each pixel that corresponds to a specific instance. Semantic segmentation refers to assigning each pixel in an image to one of a predefined set of semantic classes. These tasks serve as an indispensable tool for scene understanding, providing precise information about the scenario. Computer vision tasks lend themselves to diverse application areas including safety-critical areas like robotics (Cartucho et al., 2021), medical diagnosis (Kang & Gwak, 2019) and automated driving (Xu et al., 2023). Along with high accuracy of the models, *prediction reliability in the form of uncertainty assessment (Maag & Riedlinger, 2024) and calibrated confidence estimates (Mehrtash et al., 2019) is also highly relevant.*

Object detectors provide confidence values ("objectness") about the correctness of each predicted object, while semantic segmentation models output a softmax probability score per pixel indicating the confidence of class affiliation. Both of the former represent notions of probability for correct predictions, which also reflects the prediction uncertainty of the model. For confidence scores to be

---

*Equal contribution.

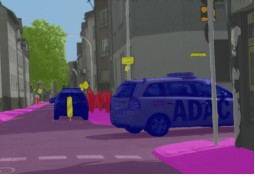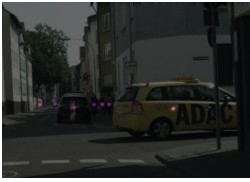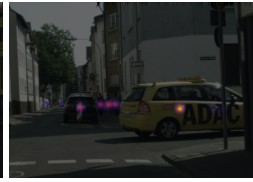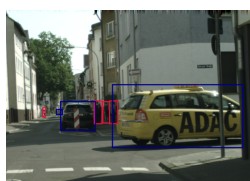

Figure 1: *Left*: Semantic segmentation prediction. *Center left*: Poisson point process intensity. *Center right*: Conditional marked Poisson point process intensity. *Right*: Bounding box prediction.

statistically reliable, the assigned confidence should match the observed prediction accuracy, e.g., out of 100 predictions with a confidence of 70% each, 70 are correct. If this is approximately the case, we call the confidence assignments "well-calibrated". However, DNN confidences are frequently miscalibrated, with confidence values not matching the observed accuracy (Küppers et al., 2022). This problem can often be tackled by confidence re-calibration methods which regularize the training objective or establish calibration by post-processing (Guo et al., 2017). Particularly in the case of object detection, the focus is mainly on correctly calibrating objectness for predicted objects, while false negatives are ignored.

The problem of miscalibration often arises from a mis-specification of the architecture and/or the loss function of the model. The latter are often created based on heuristics and practical considerations, e.g., feature pyramid networks (Lin et al., 2017) or anchor box procedures (Redmon et al., 2016), working well empirically but lacking a probabilistic foundation. *The logic of conditional stochastic independence of pixel-wise predictions given the input image, like in semantic segmentation or objectness, are violated in practice.* Moreover, the commonly used cross-entropy loss with one-hot targets does not intrinsically give rise to calibration, since its main goal is to optimize classification accuracy (Liu et al., 2024). The DNN is likely to predict the true class with high probability without calibrating the probabilities of the other classes. Thus, models often predict extremely high or low probabilities tending towards overconfidence.

Aside from the calibration of foreground predictions, object detection exhibits the peculiarity that the areas outside detections are (naturally) not annotated and learned as background during training. *Object detectors do not provide explicit uncertainty assessment for areas outside their own predictions.* That is, the model does not make a probability assessment of whether an area without detected objects is truly free of obstacles since such a mechanism is not built into the loss function. Such a model is then often overconfident in assuming that "no object" means the area is safe. This has significant impact, e.g., in trajectory planning of autonomous cars and robotic agents where it is central to assess whether the planned track actually is collision-free. Conventional object detectors are unsuitable for this purpose because they only provide discrete and potentially erroneous object predictions without having a calibrated, quantitative idea of the collision risk over a continuous trajectory, making their predictions unreliable for collision-free planning. We consider the absence of adequate methods addressing this question to be a major safety and research gap in the field of safe automated driving. Our approach addresses this flaw and constitutes a step towards safe navigation.

In this work, we propose an object detection model based on spatial statistics, regarding bounding box data as realizations of a marked point process (Møller et al., 2003; Sherman, 2011). Such models have been used to model spatial phenomena, e.g., in astronomy, epidemiology or geostatistics, to describe the probabilistic occurrence of spatial point events. We add marks to center point events describing the geometry of objects (width and height) and class affiliation obtaining an object detection model based on the logic of counting processes. When conditioned on the number of events, our model allows for likelihood-based end-to-end training. Out of the box, *our model allows to compute well-defined notions of confidence for the event that a certain region in space is drivable*. We differentiate between two notions of "being drivable" depending on whether we consider to marked or non-marked point process. This gives rise to an object detector which defines confidences that an arbitrary test region in space is truly free of objects. This model conceptually provides aleatoric uncertainty, but can be combined with methods for estimating epistemic uncertainty, such as Bayesian approximation. An exemplary prediction of our model is shown in fig. 1. We summarize our contribution as follows:

- We develop a deep learning framework for modeling the intensity function of spatial point processes based on a negative log-likelihood loss function to obtain probabilistic statements about empty space.
- We propose a mathematically principled model for object detection based on the theory of marked Poisson point processes allowing for end-to-end training. Contrasted with existing object detectors, we require zero training hyperparameters and only a single inference hyperparameter.
- We propose an evaluation protocol for testing calibration of empty space confidences and evaluate our method on street scene data, indoor robotics data as well as on drone data as practical use cases.
- The proposed model is shown to be well-calibrated on image regions while having comparable performance to standard object detection architectures.

To our knowledge, this is the first time, a spatial statistics approach has been used for deep object recognition in computer vision. Our derivation of the loss function is a principled and mathematically underpinned approach to deep object detection highlighting parallels and differences with typical design choices in object detection. The source code of our method is publicly available at `https://github.com/CMPPP-CV/cmpppnet`.

## 2  RELATED WORK

**Drivable area prediction.**   Drivable area or free space detection refers to a task related to autonomous driving that does not focus on objects, but on the classification between drivable and background (everything else). This can be done based on different sensoric data such as LiDAR, radar, camera and aerial imaging (Hortelano et al., 2023) where we focus on camera data. The basis for the drivable area prediction is often a semantic segmentation architecture, since pixel level predictions are desired (Chen et al., 2025; Qiao & Zulkernine, 2021). Different network design choices have been pursued to solve drivable area prediction including fully convolutional models (Yu et al., 2023; Chan et al., 2019), incorporation of LSTM layers (Lyu et al., 2019) or boundary attention units (Sun et al., 2019) to overcome task-specific difficulties. Another approach to improve drivable area detection involves multi-task architectures, e.g., for different segmentation tasks (Lee, 2021), detection of lane lines (Wang et al., 2024) or instance segmentation (Luo et al., 2024). In Qian et al. (2020), a unified neural network is constructed to detect drivable areas, lane lines and traffic objects using sub-task decoders to share designate influence among tasks. In addition to combining tasks, different types of sensor data can also help, such as combining RGB images with depth information (Fan et al., 2020; Jain et al., 2023). An uncertainty-aware symmetric network is presented in Chang et al. (2022) to achieve a favorable speed-accuracy trade-off by fully fusing RGB and depth data.

In contrast to our methodology, drivable area prediction and free space detection are based on design choices and practical considerations and do not exhibit well-defined notions of void confidence. In direct contrast to our approach, which *assigns occupation confidences to arbitrary test regions*, the aim of drivable area prediction is to concretely localize regions that do not contain obstacles.

**Object detection models.**   In computer vision, space occupation by objects is classically treated in the form of object detection. Object detectors typically model the existence of foreground objects by assigning an objectness score to super pixels (Farhadi & Redmon, 2018; Ren et al., 2015; Liu et al., 2016) which is trained via binary cross entropy. Super pixels with sufficient objectness then contribute a bounding box prediction to the final set of detections which is subsequently pruned by non-maximum-suppression to filter double predictions that indicate the same object. Our proposed intensity model shares similarities with objectness in single stage detectors like the YOLO models (Redmon et al., 2016; Redmon & Farhadi, 2017; Farhadi & Redmon, 2018), SSD (Liu et al., 2016), FCOS (Tian et al., 2019) or also CenterNet (Duan et al., 2019) in that a spatial feature map is computed which indicates the existence of foreground objects. The same mechanism is used in region proposal modules of two-stage architectures like the R-CNN family (Girshick et al., 2014; Ren et al., 2015; Girshick, 2015; Cai & Vasconcelos, 2018) or probabilistic two-stage extensions like CenterNetv2 (Zhou et al., 2021). However, there is a central difference in the meaning of the feature maps. Objectness is interpreted on the basis of superpixels and in a binary fashion (foreground/background) where different pixels are stochastically independent as prescribed by the

loss function. *Intensity on the other hand is built on a cumulative logic via integration over image regions* and, hence, modeled densely on the same resolution as the input image. Spatial dependence between pixels is incorporated by a free spatial point process loss function presented in section 3 and leads to statistically reliable confidence estimation for space occupation. Due to its focus on modeling center points, anchor-free models like CenterNet (Duan et al., 2019) and FCOS (Tian et al., 2019) share similarities with our model in terms of inference logic. We emphasize, however, that the "centerness" score of CenterNet is conceptually more closely related with objectness than intensity due to the built-in superpixel independence rendering both models misspecified for calibration of space occupation. Additionally, we do not claim that our model is capable of outperforming the above-mentioned object detectors in terms of detection accuracy as our model has not gone through several generations of architectural optimization. Rather, *we present the first ever approach to object detection which is fully probabilistically founded.* Our model is *capable of assigning calibrated void/occupation probabilities to arbitrary regions in space*, a central research gap for autonomous driving and robotic perception. An important evaluation protocol for object detectors in terms of the PDQ score (Hall et al., 2020) has been introduced viewing object detection in light of the underlying probabilistic modeling assumptions for bounding box regression. Bounding box localization is modeled by Gaussian distributions, however, irrespective of the chosen optimization objective. In contrast, we rigorously derive the correct correspondence between regression loss and probabilistic model for bounding box uncertainty.

## 3 METHOD DESCRIPTION

In this section, we propose our object detection model architecture and inference logic. To this end, we derive the optimization functional from the theory of spatial point processes and propose an implementation modeling point process intensity coupled with bounding box properties as marks.

### 3.1 MARKED POINT PROCESS MODELS

**Bounding box data as marked point configurations.** We briefly introduce marked point processes as a way of representing object detection. A statistical model has to reflect the structure of the data. Intuitively, in object detection *each foreground object corresponds to a (center-)point equipped with a mark describing size and class*. Each instance is represented as a tuple $z = (\boldsymbol{\xi}, m)$, where $\boldsymbol{\xi} = (\xi_x, \xi_y) \in [0,1]^2$ encodes the location of the instance's center while $m = (h, w, \kappa) \in \mathbb{R}^2 \times \mathcal{C}$ provides height and width of the bounding box as well as the label $\kappa$ from a discrete set of classes $\mathcal{C}$. We call $\boldsymbol{\xi}$ the *(center) point location* and $m$ the *mark* attached to $\boldsymbol{\xi}$.

The object detection ground truth data for a given image consists of a set $\{z_1, \ldots, z_n\}$ of marked points, where $n \in \mathbb{N}_0$ varies between images. The corresponding point configuration $\{\boldsymbol{\xi}_1, \ldots, \boldsymbol{\xi}_n\}$ lives in the space of $n$-point configurations $\Xi_n$ over $[0,1]^2$, where we allow for zero points, i.e., $\Xi_0 = \{\emptyset\}$, containing only the "empty configuration". The center points on a generic image are then specified by an element of $\Gamma = \bigoplus_{n \in \mathbb{N}_0} \Xi_n$, the space of finite point patterns. Let $(\Omega, \mathcal{A}, \mathbb{P})$ be a probability space, then *a random variable $X : \Omega \to \Gamma$ is said to be distributed according to a (spatial) point process over $[0,1]^2$*. $N(A) = |X \cap A|$ is the associated counting process counting the number of points $X(\omega) = (\boldsymbol{\xi}_1(\omega), \ldots, \boldsymbol{\xi}_{N(\omega)}(\omega))$ inside the measurable set $A \subseteq [0,1]^2$. $N(\omega) = N([0,1]^2)(\omega)$ gives the total count of points for the given random parameter $\omega \in \Omega$. Likewise, *a marked point process $X_M : \Omega \to \Gamma_M$ takes values in $\Gamma_M = \bigoplus_{n \in \mathbb{N}_0} (\Xi_n \times M^n)$, where the mark space $M$ for object detection is given by $M = \mathbb{R}^2 \times \mathcal{C}$*. The projection $\{z_1, \ldots, z_n\} \mapsto \{\boldsymbol{\xi}_1, \ldots, \boldsymbol{\xi}_n\}$ associates a (non-marked) point process $X$ with $X_M$ and $X_M$ can be considered as a point process over the extended space $[0,1]^2 \times M$.

We are interested in statistical models that are eligible to model the distribution of a marked point process $X_M$. Here we use the Poisson point process (PPP) as the simplest model and "workhorse" of spatial statistics. It is based on an intensity measure $\Lambda = \Lambda(z) \, \mathrm{d}z$, where $\mathrm{d}z$ is the Lebesgue measure on $[0,1]^2 \times \mathbb{R}^2 \times \mathcal{C}$ where on $\mathcal{C}$ we chose the cardinality measure. For $A_M \subseteq [0,1]^2 \times M$ specify the statistics of $N_M(A_M)$ by the Poisson distribution with intensity $\Lambda(A_M) = \int_{A_M} \Lambda(z) \, \mathrm{d}z$, i.e., the probability of finding $n \in \mathbb{N}_0$ marked point instances in $A_M$ is given by

$$\mathbb{P}\left(N_M(A_M) = n\right) = \tfrac{1}{n!}\Lambda(A_M)^n \cdot \exp\left(-\Lambda(A_M)\right). \tag{1}$$

Choosing $A_M = A \times M$, $A \subseteq [0,1]^2$ measurable, also the associated point process of center points $N(A)$ is a Poisson point process with intensity on $[0,1]^2$ given by

$$\lambda(\boldsymbol{\xi}) = \int_M \Lambda(\boldsymbol{\xi}, m)\, \mathrm{d}m. \tag{2}$$

**Object detection via conditional marked Poisson point processes.** Equation (1) defines a probabilistic model for bounding box data $\{z_1, \ldots, z_n\}$ on a fixed image $I$. *A predictive model can be derived by conditioning the intensity $\Lambda(z) = \Lambda(z|I)$ on the input $I \in [0,1]^2 \times \mathbb{R}^3$ represented by three values for RGB channels.* We call such a model a conditional marked Poisson point process (CMPPP) model. The input resolution of the pixelized image $I_{\mathrm{d}}$ determines the discretization of the Lebesgue measure over $[0,1]^2$ and, therefore, the area/mass of each pixel. We denote by $\Pi$ the set of pixel locations in $[0,1]^2$ associated with $I_{\mathrm{d}}$.

Let us now consider suitable models for object detection from CMPPP. We take a factorizing model

$$\Lambda(z|I) = \Lambda(\boldsymbol{\xi}, m|I) = \widetilde{\lambda}(\boldsymbol{\xi}|I) \cdot p(m|\boldsymbol{\xi}, I), \tag{3}$$

where $\Lambda(z|I) \geq 0$ and $p(m|\boldsymbol{\xi}, I)$ is a Markov kernel that models the probability density of the mark $m = (h, w, \kappa)$ given that there is a bounding box centered in $\boldsymbol{\xi}$. Since $\int_M p(m|\boldsymbol{\xi}, I)\, \mathrm{d}m = 1$ for any $\boldsymbol{\xi} \in [0,1]^2$, $\widetilde{\lambda}(\boldsymbol{\xi}) = \lambda(\boldsymbol{\xi})$ follows from eq. (2) and eq. (3). It remains to model the conditional intensity and the conditional probability on mark space as follows

$$\lambda(\boldsymbol{\xi}|I_{\mathrm{d}}) = \exp\left(L_{\boldsymbol{\xi}}(I_{\mathrm{d}})\right), \qquad p(m|\boldsymbol{\xi}, I_{\mathrm{d}}) = p_{w,h}(B_{\boldsymbol{\xi}}(I_{\mathrm{d}})) \cdot p_\kappa(C_{\boldsymbol{\xi}}(I_{\mathrm{d}})|B_{\boldsymbol{\xi}}(I_{\mathrm{d}})) \tag{4}$$

for a continuous distribution $p_{w,h}$ for the bounding box width and height and some categorical distribution $p_\kappa$ for the object class. Note, that both, $p_{w,h}$ and $p_\kappa$, are technically conditional on $I_{\mathrm{d}}$, which is omitted in the notation for readability. We use an exponential parametrization to ensure non-negativity of the intensity. *The functions $L$, $B$ and $C$ are realized as dense (pixel-wise, i.e., indexed by pixel $\boldsymbol{\xi}$) output of a neural network* which is fitted based on data $(I_d, z_1, \ldots, z_n)$, where $z_i = (\boldsymbol{\xi}_i, w_i, h_i, \kappa_i)$ for $i = 1, \ldots, n$ in a maximum-likelihood approach. In our implementation, $p_{w,h}$ is realized as a bivariate (independent) Laplace distribution with location parameter $(w, h)$ and isotropic scale parameter $\sigma$ which is discussed further in the next section. We model $p_\kappa$ by a softmax distribution with logits $C_{\boldsymbol{\xi}}(I_{\mathrm{d}})$, independently of $B_{\boldsymbol{\xi}}(I_{\mathrm{d}})$.

## 3.2 Likelihood training approach and inference for CMPPP models

**CMPPP loss function.** Commonly, one chooses the negative log-density function with respect to the Lebesgue measure $\mathrm{d}x$ as loss function for a parametric model density $p_\theta$, i.e., $\ell(x, \theta) = -\log p_\theta(x)$ where $X \sim \mu_\theta = p_\theta(x)\, \mathrm{d}x$ is a hypothesis on the distribution of the data represented by (independent copies of) the random variable $X : \Omega \to \mathbb{R}^d$. This is no longer feasible in the setting of point processes $X : \Omega \to \Gamma$ as no Lebesgue measure exists on the infinite dimensional space $\Gamma$ and therefore the notion of a density w.r.t. $\mathrm{d}x$ is ill-defined. The notion of likelihood can, however, be adapted to reference measures other than $\mathrm{d}x$. Let, in the finite dimensional setting, $\mu$ be another (probability) measure on $\mathbb{R}^d$ with density $p_\mu(x)$ such that $p_\mu(x) = 0$ implies $p_\theta(x) = 0$. We obtain

$$\ell(x, \theta) = -\log p_\theta(x) = -\log \frac{p_\theta(x)}{p_\mu(x)} - \log p_\mu(x) = -\log\left(\frac{\mathrm{d}\mu_\theta}{\mathrm{d}\mu}(x)\right) - \log p_\mu(x). \tag{5}$$

Here $\frac{\mathrm{d}\mu_\theta}{\mathrm{d}\mu}(x) = \frac{p_\theta(x)}{p_\mu(x)}$ is the Radon-Nikodym (RN) derivative of $\mu_\theta$ with respect to the reference measure $\mu$. As the gradients $\nabla_\theta \ell(x, \theta)$ of eq. (5) do not depend on $p_\mu$, training with the negative log likelihood loss is equivalent to training with the negative log-RN derivative. Furthermore, the gradients also do not depend on the particular choice of $\mu$, provided the RN derivative exists.

The training of stochastic models with an infinite dimensional state space is based on the insight that RN derivatives with respect to adequately chosen reference measures $\mu$ still exist, as long as $\mathrm{d}\mu_\theta = p_\theta\, \mathrm{d}\mu$ holds, where the relative density $p_\theta(x) = \frac{\mathrm{d}\mu_\theta}{\mathrm{d}\mu}(x)$ again is the Radon-Nikodym derivative at $x \in \Gamma$. *As the reference measure, we choose the distribution $\mu$ of the homogeneous PPP over $[0,1]^2$ with intensity function $\lambda_\mu \equiv 1$ (the normalized Lebesgue measure).* The RN derivative of the distribution $\mu_\theta$ of a process with intensity $\lambda_\theta(\boldsymbol{\xi})$ w.r.t. $\mu$ then is

$$\frac{\mathrm{d}\mu_\theta}{\mathrm{d}\mu}(x) = \exp\left(-\int_{[0,1]^2}(\lambda_\theta(\boldsymbol{\xi}) - 1)\, \mathrm{d}\xi\right) \cdot \prod_{l=1}^n \lambda_\theta(\boldsymbol{\xi}_l), \quad x = \{\boldsymbol{\xi}_1, \ldots, \boldsymbol{\xi}_n\} \in \Gamma. \tag{6}$$

This identity is a standard exercise in textbooks on spatial statistics, see, e.g., (Daley & Vere-Jones, 2006). For the convenience of the reader we provide a proof in appendix A.1. Equation (6) immediately generalizes to marked CMPPP if we replace $x \in \Gamma$ with $x = \{z_1, \ldots, z_n\} \in \Gamma_M$, $\lambda_\theta(\boldsymbol{\xi}_i)$ with $\Lambda_\theta(z_i|I)$, $\boldsymbol{\xi}$ with $z$, $[0,1]^2$ with $[0,1]^2 \times M$ and $d\xi$ with $dz$.

Inserting eq. (3) and eq. (4) into the updated version of eq. (6) and taking the negative logarithm, we obtain the CMPPP loss function for $(L^\theta, B^\theta, C^\theta)$:

$$\ell(x, \theta) = \int_{[0,1]^2} e^{L_\xi^\theta(I_d)} \, d\xi - \sum_{i=1}^n L_{\boldsymbol{\xi}_i}^\theta(I_d) - \sum_{i=1}^n \left[ \log(p_{w,h}(B_{\boldsymbol{\xi}_i}^\theta(I_d))) + \log(p_\kappa(C_{\boldsymbol{\xi}_i}^\theta(I_d))) \right]. \tag{7}$$

Here, $z = \{z_1, \ldots, z_n\} \in \Gamma_M$ is the bounding box configuration observed in the image $I$. Minimizing the loss function now enables a strictly likelihood-based training.

It is now easy to interpret the first two terms of eq. (7) as the loss for the center point intensity $\lambda_\theta = \exp(L^\theta)$ and hence a loss for a "distributed objectness score". Assuming a Laplace distribution, the $p_{w,h}$-term yields the standard $L^1$-loss for bounding box regression and the $p_\kappa$-term the cross entropy classification loss. The integral in the first term is discretized over $[0,1]^2$ according to the image resolution $H \times W$ as

$$\int_{[0,1]^2} \exp\left( L_\boldsymbol{\xi}^\theta(I_d) \right) d\xi \approx \sum_{\boldsymbol{\xi} \in \Pi} \exp\left( L_\boldsymbol{\xi}^\theta(I_d) \right) \cdot \frac{1}{HW} \tag{8}$$

with each pixel obtaining area $\frac{1}{|\Pi|} = \frac{1}{HW}$. We note that we choose to not fit the scale variable $\sigma$ of the Laplace distribution by the model. This makes the training of $B^\theta$ is independent of the value of $\sigma > 0$ such that training with the $L^1$-loss can be conducted first resulting in the approximately optimal weights $\widehat{\theta}$. Thereafter, the maximum likelihood equations for $\sigma$ yield $\widehat{\sigma} = \frac{1}{n} \sum_{i=1}^n \left\| \binom{w_i}{h_i} - B_{\boldsymbol{\xi}_i}^{\widehat{\theta}}(I_d) \right\|_1$, i.e., the mean average deviation of the bounding box regression *allowing for object detection training with zero hyperparameters*.

**Probabilistic predictions of empty space.** On the basis of a trained model with parameters $(\widehat{\theta}, \widehat{\sigma})$ we now derive a probabilistic conditional prediction that a measurable test region $A \subseteq [0,1]^2$ is "free of objects". We interpret this statement in two different ways, i.e., we consider two notions of emptiness.

On the one hand, it may mean "*A is free of object centers*". The random variable $X$ models object centers and $N = \delta_X$ is the associated counting (Dirac-) measure. Then, the event that "$A$ is free" amounts to $X \cap A = \emptyset$ or $N(A) = 0$. Using the Poisson statistic (1) for the associated center point process, i.e., $\lambda_{\widehat{\theta}}(\cdot|I_d) = \exp(L_{(\cdot)}^{\widehat{\theta}})$ instead of $\Lambda(\cdot)$, we obtain

$$\mathbb{P}_{\widehat{\theta}}(N(A) = 0|I) = \exp\left( -\int_A \lambda_{\widehat{\theta}}(\boldsymbol{\xi}|I) \, d\xi \right) \approx \exp\left( -\frac{1}{HW} \sum_{\boldsymbol{\xi} \in A \cap \Pi} \exp\left( L_\boldsymbol{\xi}^{\widehat{\theta}}(I_d) \right) \right). \tag{9}$$

On the other hand, an interpretation is the event that "*A does not intersect any of the bounding boxes*" $[\xi_x - \frac{w}{2}, \xi_x + \frac{w}{2}] \times [\xi_y - \frac{h}{2}, \xi_y + \frac{h}{2}] =: b(z)$ for any $z = (\xi_x, \xi_y, w, h, \kappa) \in X_M$. To this end, consider the critical set $D^c(\boldsymbol{\xi}) \subseteq [0,1]^2 \times M$ for a point $\boldsymbol{\xi} \in A$, that is, the set of all bounding boxes $z' \in [0,1]^2 \times M$ such that $\boldsymbol{\xi}$ is contained in the corresponding bounding box $b(z')$. It is easily seen that $D^c(\boldsymbol{\xi}) = \{z' = (\boldsymbol{\xi}', w', h', \kappa') \in [0,1]^2 \times M : |\xi_x - \xi_x'| \leq w/2 \text{ and } |\xi_y - \xi_y'| \leq h/2\}$. The critical set for the entire region $A$ then is given by $D^c(A) = \bigcup_{\boldsymbol{\xi} \in A} D^c(\xi)$ and the probability that no bounding box in a given image $I$ intersects $A$ is

$$\mathbb{P}_{\widehat{\theta}}(N(D^c(A)) = 0|I) = \exp\left( -\int_{D^c(A)} \Lambda_{\widehat{\theta}}(z|I) \, dz \right), \tag{10}$$

where $\Lambda(z|I)$ is evaluated using eq. (3) and eq. (4). Let us shortly consider the evaluation of the integral on the right hand side of eq. (10). By our modeling ansatz, the integral over $\Lambda_{\widehat{\theta}}$ separates to

$$\int_{[0,1]^2 \setminus A} \lambda_{\widehat{\theta}}(\xi|I) \cdot \int_{\{m \in M : b(\xi, m) \cap A \neq \emptyset\}} p(m|\xi, I) \, dm \, d\xi. \tag{11}$$

For the special case that the test region $A$ is a rectangle with center point $(\xi_x^A, \xi_y^A) \in [0,1]^2$, width $w^A$ and height $h^A$, $A$ intersects $b(\boldsymbol{\xi}, w, h, \kappa)$ if both, $|\xi_x^A - \xi_x| \leq \frac{1}{2}(w^A + w)$ and $|\xi_y^A - \xi_y| \leq \frac{1}{2}(h^A + h)$ hold. The inner integral then factorizes and the integral in eq. (10) becomes

$$\sum_{\boldsymbol{\xi} \in \Pi \setminus A} e^{L_\xi^{\widehat{\theta}}(I_d)} \cdot \int_{2|\xi_x^A - \xi_x| - w^A}^{\infty} \frac{1}{2\sigma} e^{-\frac{1}{\sigma}|w - B_{\boldsymbol{\xi},w}^{\widehat{\theta}}(I_d)|} dw \cdot \int_{2|\xi_y^A - \xi_y| - h^A}^{\infty} \frac{1}{2\sigma} e^{-\frac{1}{\sigma}|h - B_{\boldsymbol{\xi},h}^{\widehat{\theta}}(I_d)|} dh \tag{12}$$

which can easily be expressed by the cumulative distribution function of $p_{w,h}$. The evaluation of the void confidence generalizes e.g., to modeling with normal distributions and can easily be algorithmically implemented. While the inner integral over $\{m \in M : b(\boldsymbol{\xi}, m) \cap A \neq \emptyset\}$ may in general be computed by logical querying of pixels and CDFs, more general shapes may also be treated via the inclusion-exclusion principle.

**Prediction of foreground objects.** For bounding box prediction, instead of the standard non-maximum suppression algorithm, *we exploit the counting statistics of the spatial point process and determine the expected number of center points in* $[0,1]^2$ *by* $\mathbb{E}[N] = \int_{[0,1]^2} \lambda_{\widehat{\theta}}(\boldsymbol{\xi}|I)\mathrm{d}\xi \approx \frac{1}{HW} \sum_{\boldsymbol{\xi} \in \Pi} \lambda_{\widehat{\theta}}(\boldsymbol{\xi}|I_\mathrm{d})$. Afterwards, we extract this number of peaks (see fig. 2) from the intensity function to find the predicted center points. As the intensity function is sharply peaked but still somewhat spread-out, we crop square patches of $32 \times 32$ pixels around determined maxima before searching for the next peak. An ablation study on crop square size, *our only hyperparameter*, is provided in appendix A.2. We observe robust behavior over large range of settings. Marks are determined by evaluation of $B^{\widehat{\theta}}(I_\mathrm{d})$ and $C^{\widehat{\theta}}(I_\mathrm{d})$ feature maps at the respective locations.

# 4 EXPERIMENTS

## 4.1 MODEL DESIGN AND EXPERIMENTAL SETTING

**Network architecture.** In order to model the functions $L$, $B$ and $C$ in eq. (4), we choose deep neural networks that are capable to compute pixel-wise outputs, in particular we utilize architectures used in semantic segmentation with $1 + 2 + |\mathcal{C}|$ output channels modeling the tuple $(L^\theta, B^\theta, C^\theta)$. Given ground truth data $(I_\mathrm{d}, z_1, \ldots, z_n)$, this allows for computing the full CMPPP loss (eq. (7)) and training end-to-end. In appendix A.5, we present additional experiments with a two-stage architecture and additional experiments where we model the residuals as normal distributions instead of Laplace distributions. We implement our CMPPP model in the MMDetection environment (Chen et al., 2019), importing segmentation architectures from MMSegmentation. Our investigations involve a DeepLabv3+ (Chen et al., 2018) model with ResNet-50 backbone, an FCN model with HRNet (Wang et al., 2021) backbone, as well as SegFormer-B5 (Xie et al., 2021) model. Training ran on a Nvidia A100 GPU with 80GBs of memory and standard (pre-set) parameter settings for training on the various datasets.

**Datasets.** In our experiments, we use three datasets: Cityscapes and TiROD, street and indoor robotics datasets where empty spaces are relevant for safe and collision-free navigation as well, as VisDrone, an aerial drone dataset where the detection of free landing sites is of interest. The Cityscapes dataset (Cordts et al., 2016) depicts dense urban traffic scenarios in various German cities. This dataset consists of 2,975 training images and 500 validation images of size $1{,}024 \times 2{,}048$ from 18 and 3 different cities, respectively, with labels for semantic and instance segmentation of road users (different vehicles and humans). From the instance labels, we obtain the center points and bounding box ground truth of the objects. The VisDrone-DET dataset (Zhu et al., 2022) shows aerial images of urban scenarios in different Chinese cities. The dataset consists of 6,471 training and 548 validation images of varying resolutions and bounding box annotations for 10 different road-related semantic classes. The Tiny Robotics Object Detection (TiROD) dataset (Pasti et al., 2025) contains frog perspective scenes recorded in indoor lab/office settings and outdoor garden settings with two different illumination conditions in 5,336 training and 666 validation images. The $640 \times 480$ pixel images contain 13 different semantic classes of foreground instances.

## 4.2 NUMERICAL RESULTS

**PPP intensity calibration.** In this section, we compare our intensity prediction of the (non-marked) PPP with an analogous semantic segmentation prediction regarding the respective calibration of predicting empty spaces. *We aim to answer the question whether a test region $A$ is drivable.* Given an input image $I_\mathrm{d}$ and trained model, a semantic segmentation model computes a probability distribution $p(\cdot|I_\mathrm{d})_{\boldsymbol{\xi}} \in [0,1]$ over $\mathcal{C}$ for each pixel $\boldsymbol{\xi} \in \Pi$ specifying the probability $p(\kappa|I_\mathrm{d})_{\boldsymbol{\xi}} \in [0,1]$ for each class $\kappa \in \mathcal{C}$. The probability that a region is drivable is given by $\mathbb{P}_S(A \text{ is drivable}) = \prod_{\boldsymbol{\xi} \in \Pi \cap A} p(\text{``road''}|I_\mathrm{d})_{\boldsymbol{\xi}}$, as the pixel predictions are assumed to be independent when conditioned

Table 1: Calibration values of semantic segmentation model ($\text{ECE}_S \downarrow$) and our PPP method ($\text{ECE}_P \downarrow$) for the Cityscapes dataset and different box sizes $s$.

| | $s$ | 250 | 500 | 750 | 1,000 | 1,500 | 2,500 | 5,000 | 10,000 |
|---|---|---|---|---|---|---|---|---|---|
| Deep-Labv3+ | $\text{ECE}_S$ | 0.1102 | 0.1741 | 0.2033 | 0.2245 | 0.2521 | 0.2667 | 0.2417 | 0.1948 |
| | $\text{ECE}_P$ | 0.0012 | 0.0017 | 0.0029 | 0.0029 | 0.0046 | 0.0062 | 0.0109 | 0.0164 |
| HRNet | $\text{ECE}_S$ | 0.0413 | 0.0859 | 0.1142 | 0.1443 | 0.1785 | 0.2206 | 0.2295 | 0.1939 |
| | $\text{ECE}_P$ | 0.0008 | 0.0012 | **0.0014** | 0.0019 | **0.0022** | **0.0041** | **0.0053** | **0.0071** |
| SegFormer | $\text{ECE}_S$ | 0.0621 | 0.0585 | 0.0609 | 0.0593 | 0.0588 | 0.0582 | 0.0626 | 0.0713 |
| | $\text{ECE}_P$ | **0.0006** | **0.0008** | **0.0014** | **0.0018** | 0.0027 | 0.0046 | **0.0053** | 0.0082 |

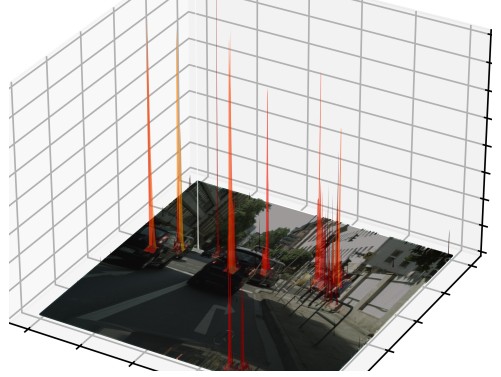

Figure 2: Intensity landscape over an input image from the Cityscapes val dataset. Peaks are mostly sharply localized and indicate foreground detections.

Figure 3: Confidence calibration plots for semantic segmentation (blue) and PPP (orange) with corresponding ECE for the Cityscapes dataset and the DeepLabv3+ detector and $s = 1,000$.

on a fixed image $I$. We consider a region to be drivable if it contains only the "road" class. For the PPP model, the probability that the region is free is derived from the case $n = 0$ in eq. (1) under the discretization (8)

$$\mathbb{P}_P(A \text{ is drivable}) = \exp\left(-\int_A \lambda(\boldsymbol{\xi})\, \mathrm{d}\xi\right) \approx \exp\left(-\tfrac{1}{|\Pi|} \sum_{\boldsymbol{\xi} \in \Pi \cap A} \lambda(\boldsymbol{\xi})\right). \tag{13}$$

We consider a region to be drivable if it contains no center point. To determine the calibration of the methods, we sample random boxes (test regions) of a fixed area $s$ (height and width chosen randomly), where the number of boxes per image is fixed (here 50). For each box, we determine whether it is drivable and determine the respective probability. Based on these quantities, we calculate the expected calibration error (ECE) (Naeini et al., 2015) to evaluate calibration.

The results for the Cityscapes dataset are shown in Table 1, depending on the test box size. We observe that calibration for smaller boxes also achieves smaller ECE errors. Furthermore, we find that HRNet is more accurately calibrated than DeepLabv3+ for semantic segmentation. The reason for this could be the difference in model capacity and that HRNet does not use significant downsampling or pyramid architecture, instead relying on high-resolution representations through the whole process. However, both convolutional networks are less well-calibrated than the SegFormer. These differences between the models become significantly smaller with our PPP method; only slight trends can still be identified. It is quite evident that our PPP method is significantly better calibrated than the semantic segmentation prediction. A corresponding confidence calibration plot is shown in fig. 3. The semantic segmentation model is consistently underconfident. If any pixel indicates that the test area may not be drivable, the confidence vanishes as expected from the probability factorization. *In comparison, our method fluctuates close to optimal calibration (gray diagonal in the plot).* We also observe good calibration on the TiROD dataset and the more complex VisDrone dataset, as shown in table 2 (top row, respectively). Moreover, we provide calibration results for re-calibrated (e.g. temperature scaled) semantic segmentation models in appendix A.3, which still remain at least an order of magnitude worse calibrated than our model. Re-calibration methods are incapable of rectifying void confidences of models which treat pixels as independent, even when fixing the test box sizes.

Table 2: Calibration values of our DeepLabv3+ PPP model ($ECE_P \downarrow$) and CMPPP object detector ($ECE_{BB} \downarrow$) for the VisDrone as well as the TiROD dataset for different box sizes $s$.

|  |  | 250 | 500 | 750 | 1,000 | 1,500 | 2,500 | 5,000 |
|---|---|---|---|---|---|---|---|---|
| VisDrone | $ECE_P$ | 0.0025 | 0.0063 | 0.0081 | 0.0090 | 0.0163 | 0.0210 | 0.1114 |
|  | $ECE_{BB}$ | 0.2344 | 0.2018 | 0.1763 | 0.1532 | 0.1199 | 0.0823 | 0.1246 |
| TiROD | $ECE_P$ | 0.0015 | 0.0046 | 0.0062 | 0.0087 | 0.0122 | 0.0198 | 0.0383 |
|  | $ECE_{BB}$ | 0.0942 | 0.1051 | 0.1120 | 0.1197 | 0.1339 | 0.1571 | 0.1783 |

Table 3: Calibration values of our CMPPP object detection models ($ECE_{BB} \downarrow$) for the Cityscapes dataset and different box sizes $s$.

|  | 250 | 500 | 750 | 1,000 | 1,500 | 2,500 | 5,000 | 10,000 |
|---|---|---|---|---|---|---|---|---|
| DeepLabv3+ | 0.1011 | 0.0925 | 0.0869 | 0.0842 | 0.0803 | 0.0697 | 0.0692 | 0.0762 |
| HRNet | 0.1223 | 0.1133 | 0.1089 | 0.1050 | 0.0954 | 0.0781 | 0.0665 | 0.0641 |
| SegFormer | 0.0905 | 0.0813 | 0.0760 | 0.0737 | 0.0627 | 0.0538 | 0.0476 | 0.0600 |

**Bounding box detection.** Using our CMPPP model, instead of the center point of an object, we receive a bounding box prediction with the corresponding class and score value. The latter reflects the probability (12) that a region is drivable, whereby we use the Laplace distribution function as in eq. (12) due to training with the $L^1$-loss. An example of CMPPP confidence and the bounding box prediction is shown in fig. 1 (see appendix A.6 for further visualizations.[1]). We observe the differences in intensity between the objects more clearly than with the PPP because box height and width contribute to the confidence level.

In this object detection application, we consider a test regions to be drivable if the test box and the ground truth bounding box do not overlap. In the same way as before, we sample 50 random test boxes of a fixed size $s$ per image to evaluate the calibration. The results depending on $s$ for Cityscapes are given in table 3 and for VisDrone and TiROD in table 2 (bottom row in each case). While the PPP only predicts the center points of objects, CMPPP adds the prediction of the height and width of the bounding box, which is more challenging for confidence calibration. We achieve slightly worse values compared to the PPP predictions, although the calibration errors are still small. The Segformer model consistently exhibits better calibration than its convolutional counterparts. However, per architecture we do not identify a clear trend between error and box size indicating robustness across test region scales.

Object detection is usually evaluated using mean average precision (mAP), which assesses detection capability and accuracy. On the two classes, persons and vehicles, the DeepLabv3+ CMPPP model achieves a mAP of 49.43%, HRNet 55.49% and SegFormer 51.04% on the Cityscapes images. In comparison, Faster R-CNN and CenterNet, well-known object detection networks, obtain mAP value of 59.32% and 57.08%, respectively. We do not claim that our model is capable of outperforming these models in terms of object detection performance as our model has not gone through several generations of architectural optimization. Rather, our model is capable of assigning well-calibrated occupation probabilities to arbitrary regions in space. Treating superpixel objectness similarly to softmax confidences, Faster R-CNN and CenterNet both compute highly ill-calibrated void confidences, obtaining ECE values of 0.9915 (for $s = 1,000$), as expected.

**Runtime and scalability.** Our models essentially have the complexity of modern semantic segmentation architectures with minimal additional post-processing. Our DeepLabv3+ model (43.6M params, 16.2 FPS) is slightly slower than a comparable Faster R-CNN (41.4M params, 29.4 FPS) while our HRNet model (65.9M params, 15.4FPS) is on par with a Faster R-CNN with ResNeSt50 backbone (65.8M params, 16.4FPS). Overall, we conclude that our model scales well along with existing architectures even without specific tuning for efficiency.

---

[1]Public demo video: https://www.youtube.com/watch?v=1zS1ajNSN-E.

## 5 LIMITATIONS

In fig. 3, we observe residual miscalibration of the model for center bins which we hypothesize is due to the fact that the model allocates significant model capacity to also calibrate $\mathbb{P}(N(A) = n|I)$ for other $n$. Our model assigns square patch intensity to any found peak during inference which often conflicts with large foreground objects whose intensity is spread over larger areas. This suggests using depth-dependent patch sizes to differentiate between objects from different size scales. Finally, road participants and obstacles occupy physical space while we model "free"/non-interacting point configurations in our model, not incorporating repelling potentials between events.

## 6 CONCLUSION

In this work, we have introduced a novel object detection architecture and learning objective guided by the question "With what probability is some particular region of the input image devoid of objects, i.e., drivable?". Following a principled approach based on the theory of spatial point processes, we have derived an object detection model which may be trained by a notion of negative log-likelihood to model the object intensity function over the input image. Modeling the point configuration with shape and class distribution markings constitutes an object detection model capable of assigning a meaningful confidence to the event of a test region intersecting any predicted object in the image. We investigate three instances of our model on three application-driven dataset and show in numerical experiments that it is capable of solid object detection performance and is well-calibrated on emptiness. Compared to semantic segmentation and conventional object detectors, we obtain significantly better confidence calibration, and particularly, the first object detection models providing reliable information about object-free areas.

## ACKNOWLEDGMENT

Funding by the Federal Ministry for Economic Affairs and Energy (BMWE) through the Safe AI Engineering consortium grant no. 19A24000U is gratefully acknowledged.

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

# A  APPENDIX / SUPPLEMENTAL MATERIAL

## A.1  COMPUTATION OF THE RADON NIKODYM DERIVATIVE AND NEGATIVE LOG-LIKELIHOOD

In this appendix we provide the computations which lead to eqs. (6) and (7) for the convenience of the reader. For the ease of notation, we perform this computation for the non-marked point process and omit conditioning on the image $I$.

**Derivation of eq. (6).**  Let thus $\mu$ be the standard (homogeneous) PPP on $\Gamma$ with intensity $\lambda_{\mathrm{hom}} \equiv 1$ with counting measure $N_{\mathrm{hom}}(A) \sim \mathrm{Poi}(\lambda_{\mathrm{hom}}(A))$ for Borel-measurable $A \in \mathcal{B}([0,1]^2)$. Further, let $\mu_\theta$ be a PPP on $\Gamma$ with intensity function $\lambda_\theta : [0,1]^2 \to [0,\infty]$ and denote by $\Lambda_\theta(A) = \int_A \lambda_\theta(\boldsymbol{\xi})\,\mathrm{d}\xi$. Analogous to eq. (1), the counting measure $N_\theta$ of $\mu_\theta$ follows the Poisson statistics

$$\mathbb{P}(N_\theta(A) = n) = \tfrac{1}{n!}\Lambda_\theta(A)^n \cdot \exp(-\Lambda_\theta(A)).$$

We now derive that the expression given in eq. (6) is indeed the correct (conditional) Radon-Nikodym derivative of $\mu_\theta$ with respect to $\mu$. Note, that the given expression holds when explicitly evaluated at $x = \{\boldsymbol{\xi}_1, \ldots, \boldsymbol{\xi}_n\} \in \Gamma$ which is an element of the $n$-point sector in $\Gamma$. We can, therefore, only expect eq. (6) to hold when conditioned on $N_\theta(A) = n$ (respectively, $N_{\mathrm{hom}} = n$).

To prove eq. (6), by Bochner's theorem it is enough to check the characteristic function $\mathbb{E}_{\mu_\theta}[e^{i\tau N_\theta(A)}]$. We obtain for $\tau \in \mathbb{R}$

$$\mathbb{E}_\mu\left[e^{i\tau N_\theta(A)}\frac{\mathrm{d}\mu_\theta}{\mathrm{d}\mu}\right] = \sum_{n=0}^\infty \frac{e^{-1}}{n!}\,\mathbb{E}_\mu\left[e^{i\tau N_\theta(A)}\frac{\mathrm{d}\mu_\theta}{\mathrm{d}\mu}\,\middle|\,N_{\mathrm{hom}}([0,1]^2) = n\right]$$

$$= \sum_{n=0}^\infty \frac{e^{-1}}{n!}\,e^{-\int_{[0,1]^2}(\lambda_\theta(\boldsymbol{\xi})-1)\,\mathrm{d}\xi} \times \int_{[0,1]^{2n}} \prod_{l=1}^n \lambda_\theta(\boldsymbol{\xi}_l)e^{i\tau\sum_{l=1}^n \mathbb{1}_A(\boldsymbol{\xi}_i)}\,\mathrm{d}\xi_1\cdots\mathrm{d}\xi_n$$

$$= e^{-\int_{[0,1]^2}\lambda_\theta(\boldsymbol{\xi})\,\mathrm{d}\xi}\sum_{n=0}^\infty \frac{1}{n!}\left(\int_{[0,1]^2}\lambda_\theta(\boldsymbol{\xi})\,e^{i\tau\mathbb{1}_A(\xi)}\,\mathrm{d}\xi\right)^n$$

$$= \exp\left(\int_{[0,1]^2}\left(e^{i\tau\mathbb{1}_A(\boldsymbol{\xi})}-1\right)\lambda_\theta(\boldsymbol{\xi})\,\mathrm{d}\xi\right) = \exp\left(\int_A \lambda_\theta(\boldsymbol{\xi})\,\mathrm{d}\xi\left(e^{i\tau}-1\right)\right),$$

which is the characteristic function of the Poisson distribution with intensity parameter $\Lambda_\theta(A)$, exactly as the Poisson distribution in eq. (1).

**Derivation of eq. (7).**  Let us now consider the computation leading to the expression of the negative log likelihood. We start with the relative conditional density given in eq. (6) and compute the negative logarithm. This amounts to

$$\int_{[0,1]^2}\lambda_\theta(\boldsymbol{\xi})\,\mathrm{d}\xi - 1 - \sum_{l=1}^n \log\left(\lambda_\theta(\boldsymbol{\xi}_l)\right),$$

replacing $\lambda_\theta(\boldsymbol{\xi}_l)$ by $e^{L_{\boldsymbol{\xi}_l}^\theta(I_d)}$ and canceling the exponential function with the logarithm. This recovers the first two terms in eq. (7). Note that the constant term $-1$ is irrelevant for the optimization objective, however, it leads to a non-negative loss function which is sometimes desirable for tracking purposes.

Like in the product in eq. (6), for a marked point configuration $x = ((\boldsymbol{\xi}_1, m_1), \ldots, (\boldsymbol{\xi}_n, m_n))$, the product in the density eq. (6) for the marked PPP by eq. (3) includes additional factors $p(m_l|\boldsymbol{\xi}_l, I)$. Note that the pre-factor $e^{\int_{[0,1]^2}(\lambda_\theta(\boldsymbol{\xi})-1)\mathrm{d}\xi}$ remains unchanged when replacing $\lambda$ by $\Lambda$ as $\int_M p(m_l|\boldsymbol{\xi}_l, I_d)\mathrm{d}m_l = 1$. To evaluate the negative log-likelihood of the additional product factors, we have to insert the second equation in eq. (4) to evaluate $p(m_l|\boldsymbol{\xi}_l, I_d)$. This provides

$$\log\left(\prod_{l=1}^n p(m_i|\boldsymbol{\xi}_l, I)\right) = \sum_{l=1}^n \log\left(\frac{1}{2\sigma}e^{-\frac{1}{\sigma}\left\|\binom{w_l}{h_l}-B_{\boldsymbol{\xi}_l}(I)\right\|_1}[\texttt{softmax}\left(C_{\boldsymbol{\xi}_l}(I)\right)]_{\kappa_l}\right)$$

and following the computation rues for the logarithm, we arrive at eq. (1). Note that here we also use that $\texttt{softmax}(C)_\kappa = \frac{e^{C_\kappa}}{\sum_{\kappa'\in\mathcal{C}}e^{C_{\kappa'}}}$ and thus $\log(\texttt{softmax}(C)_\kappa) = C_\kappa - \log(\sum_{\kappa'\in\mathcal{C}}e^{C_{\kappa'}})$.

Table 4: Ablation on the box size for peak detection for the Cityscapes dataset on the DeepLabv3+ based network indicating the number of false positives (FPs ↓) and mean average precision (mAP↑).

|  | 8 | 22 | 24 | 26 | 28 | 30 | 32 | 34 | 36 | 38 | 40 | 42 | 100 |
|---|---|---|---|---|---|---|---|---|---|---|---|---|---|
| FPs | 2,305 | 1,695 | 1,682 | 1,669 | 1,663 | 1,664 | 1,648 | 1,634 | 1,632 | 1,642 | 1,662 | 1,676 | 2,374 |
| mAP | 43.46 | 48.97 | 49.06 | 49.23 | 49.27 | 49.33 | 49.43 | 49.52 | 49.57 | 49.54 | 49.40 | 49.16 | 41.89 |

## A.2 ABLATION STUDY

Our only hyperparameter is the box size for peak detection during inference. We performed a robustness check/ablation on the box size for the Cityscapes dataset on the DeepLabv3+ based network and indicate the number of false positives and the mAP in each case (see table 4). The default size of the results from the main paper was $32 \times 32$. We observe robust behavior over large range of settings.

## A.3 COMPARISON WITH RE-CALIBRATION METHODS

To measure the miscalibration, the expected calibration error (ECE) error is frequently used (Guo et al., 2017) where all samples of a dataset are binned based on their predicted confidence information to measure the accuracy in each bin, which is an approximation to the definition of calibration. On the one hand, binning techniques were developed, for example histogram binning (Zadrozny & Elkan, 2001), isotonic regression (Zadrozny & Elkan, 2002) and Bayesian binning (Naeini et al., 2015), which group all samples by their predicted confidence and assign the observed bin accuracy as a calibrated confidence to the grouped samples. On the other hand, scaling methods were considered, e.g. Platt scaling (Platt, 1999), temperature scaling (Guo et al., 2017), beta calibration (Kull et al., 2017) and Dirichlet calibration (Kull et al., 2019), which rescale the logits before sigmoid or softmax operation to obtain calibrated confidence estimates, but differ in their assumptions about the input data and the resulting recalibration scheme. Scaling methods achieve good calibration performance given only a small amount of samples compared to binning methods, but are limited by the underlying parametric assumptions. For this reason, combinations of both approaches have been developed (Ji et al., 2019; Kumar et al., 2019). Another line of work addresses calibration at inference time by explicitly estimating prediction quality by uncertainty features and using them to recalibrate object detection and instance segmentation outputs (Maag et al., 2021; Riedlinger et al., 2023). Another strategy for confidence calibration is to adjust the optimization objective during model training (Pereyra et al., 2017; Kumar et al., 2018; Mukhoti et al., 2020; Fernando & Tsokos, 2022).

Our motivation for applying spatial point process models is that void confidences from pixel classifiers such as semantic segmentation or objectness cannot be rectified by re-calibration techniques. We provide an intuitive conceptual explanation of this phenomenon by the following thought experiment. For test region $A$, a semantic segmentation or objectness model predicts some void confidence $\widehat{p}(A)$ as a product of confidences/objectness values over (super-)pixels within $A$ as described in our paper. We show that, and argue why, these values are vast under-estimations of the measured void frequencies. Increasing the area by a factor of 2 to an area $A'$ will also increase the number of factors in the confidence multiplication by 2, so we expect a behavior of $\widehat{p}(A') \approx [\widehat{p}(A)]^2$, i.e., we have exponential scaling with respect to the increasing factor of the area. One the one hand, this means that it is impossible to re-calibrate well across significantly differing test region sizes. On the other hand, even for fixed areas of test regions $A$, the confidences obtained from a pixel classifier by multiplying per-pixel confidences cannot even be re-calibrated to a satisfactory degree. This is because the model is not designed to yield meaningful confidences across image neighborhoods.

For empirical validation of this claim, we provide results for semantic segmentation and objectness post-hoc calibration which show that satisfactory re-calibration of the obtained confidences is not possible, even when the area of test regions is fixed. In table 5, the calibration values for semantic segmentation with temperature and Platt scaling are shown. We fit a temperature/Platt scaling to 20% of the Cityscapes validation data and evaluate on the complementary set. For each value of $s$, we fit and evaluate on test boxes of that size exclusively as this is a best case scenario for the re-calibration of pixel classifiers. No generalization to other (significantly larger or smaller) test box sizes is required. Realistically, calibration would be performed at the pixel level and then tested on other sizes, which yields poorer results than scaling for the respective test box size as shown in the

Table 5: Calibration values of semantic segmentation model without ($\text{ECE}_S$ ↓) and with scaling (temperature or Platt) and our PPP method ($\text{ECE}_P$ ↓) for the Cityscapes dataset and different box sizes $s$.

| | $s$ | 1 | 250 | 500 | 750 | 1,000 | 1,500 | 2,500 | 5,000 | 10,000 |
|---|---|---|---|---|---|---|---|---|---|---|
| Deep-Labv3+ | $\text{ECE}_S$ | 0.0591 | 0.1102 | 0.1741 | 0.2033 | 0.2245 | 0.2521 | 0.2667 | 0.2417 | 0.1948 |
| | $\text{ECE}_S$ ts | 0.0511 | 0.1286 | 0.1411 | 0.1525 | 0.1618 | 0.1653 | 0.1797 | 0.1837 | 0.0922 |
| | $\text{ECE}_S$ platt | 0.0176 | 0.0230 | 0.0335 | 0.0492 | 0.0544 | 0.0705 | 0.0548 | 0.0521 | 0.0507 |
| | $\text{ECE}_P$ | 0.0196 | 0.0012 | 0.0017 | 0.0029 | 0.0029 | 0.0046 | 0.0062 | 0.0109 | 0.0164 |
| HRNet | $\text{ECE}_S$ | 0.0682 | 0.0413 | 0.0859 | 0.1142 | 0.1443 | 0.1785 | 0.2206 | 0.2295 | 0.1939 |
| | $\text{ECE}_S$ ts | 0.0606 | 0.0378 | 0.0596 | 0.1245 | 0.1329 | 0.1397 | 0.1520 | 0.1597 | 0.1641 |
| | $\text{ECE}_S$ platt | 0.0203 | 0.0097 | 0.0096 | 0.0166 | 0.0212 | 0.0298 | 0.0444 | 0.0799 | 0.0988 |
| | $\text{ECE}_P$ | 0.0195 | 0.0008 | 0.0012 | **0.0014** | 0.0019 | **0.0022** | **0.0041** | **0.0053** | **0.0071** |
| Seg-Former | $\text{ECE}_S$ | 0.0755 | 0.0621 | 0.0585 | 0.0609 | 0.0593 | 0.0588 | 0.0582 | 0.0626 | 0.0713 |
| | $\text{ECE}_S$ ts | 0.0858 | 0.0434 | 0.0378 | 0.0379 | 0.0338 | 0.0344 | 0.0333 | 0.0364 | 0.0481 |
| | $\text{ECE}_S$ platt | 0.0142 | 0.0096 | 0.0113 | 0.0150 | 0.0159 | 0.0095 | 0.0133 | 0.0193 | 0.0303 |
| | $\text{ECE}_P$ | 0.0196 | **0.0006** | **0.0008** | **0.0014** | **0.0018** | 0.0027 | 0.0046 | **0.0053** | 0.0082 |

table. Although this re-scaling slightly improved void calibration of the baseline models, they remain at least an order of magnitude worse calibrated than our model.

We perform the same re-calibration scheme for our object detection baselines (Faster R-CNN and CenterNet). The ECE value for both models tested for a sample box size of $s = 1,000$ improves from $0.9915$ when uncalibrated to $0.5435$ with temperature scaling and $0.0023$ with Platt scaling. While calibrating test boxes on fixed size yields improved results, this gain stems from size-specific calibration effects. Once the calibration model is trained at one scale and evaluated on different test box sizes, performance degrades to levels comparable to the uncalibrated baseline. The problem with calibration for object detection models comes from the fact that uncertainty relates to a question and a prediction. As object detectors offers no prediction for the probability that a region is not occupied, it cannot be scaled or re-calibrated. Taking the confidence scores as a surrogate for the pixel-wise prediction, we end up in the same problem we found for semantic segmentation. On the pixel level ($s = 1$), these models are calibrated. But the more pixels a region contains, the less calibrated are pixel-based predictions. Consider a thought experiment and let the resolution of the image tend to infinity. In this limit, the only calibrated pixel-based models would be the perfect models that never err. All other perception networks that are calibrated on the pixel level will predict zero probability for empty space for any region.

## A.4 Ensembling and Random Seeds

Repeated training of the DeepLabv3+ CMPPP model on Cityscapes on 5 different random seeds reveals that model performance is reasonably robust with respect to initial seeds. We evaluate the object detection performance as in section 4.2 and obtain an mAP statistics of $51.46 \pm 0.82$ across the 5 models showing standard multi-seed variation. Additionally, we evaluate the models as a small ($n = 5$) ensemble and evaluate both, object detection performance and calibration of the ensemble. To this end, we average the computed intensity functions and the mark feature maps and perform the same prediction algorithm as for the individual CMPPP model and obtain a significantly improved mAP of $54.29$ while the ECE for test box size $s = 1,000$ remains similar to the values reported in table 3 with $0.10$. Therefore, we observe improved object detection performance under preserved calibration properties. Figure 4 shows visualizations of the intensity standard deviation of the deep ensemble. While generally, the variance is highly correlated with the mean value itself, we can see the high uncertainty mainly in areas around the centers of foreground objects. Instances that appear small in the input image due to their distance to the ego car have strongly concentrated variances while large appearing instances have intensity variance spread out over a respectively large area.

## A.5 Numerical Results for Different Model Design Choices

**Two-stage Architecture.** In addition to the proposed "one-stage" model, it may seem natural to design a two-stage architecture with a first stage $\theta_L$ modeling a region proposal in the shape of a

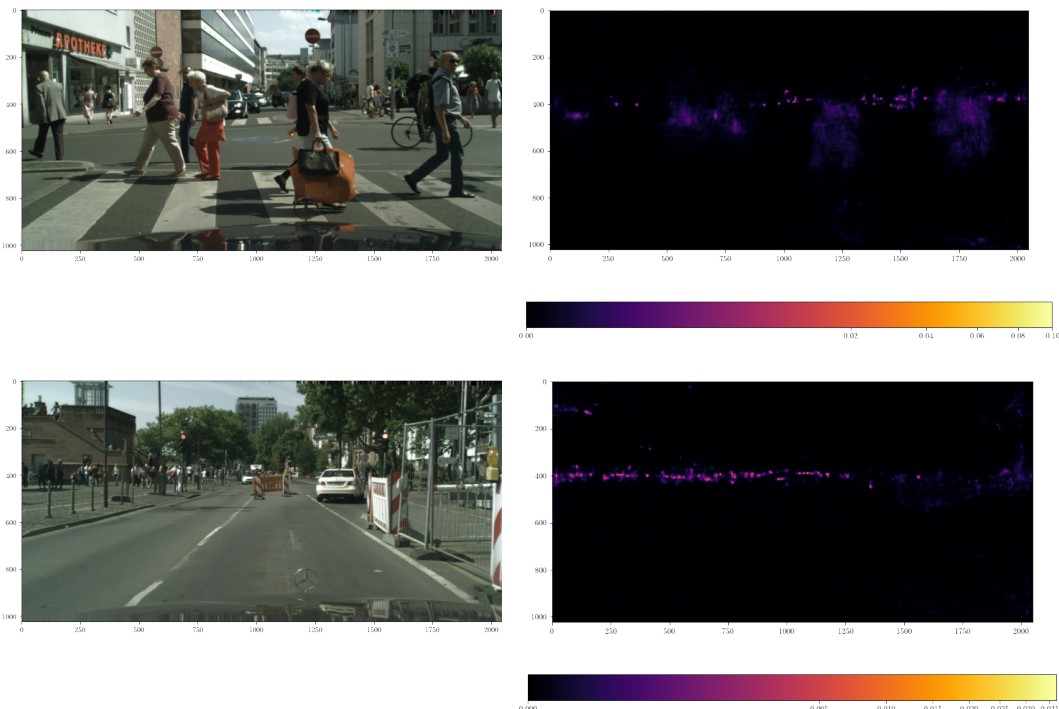

Figure 4: Visualization of ensemble intensity variance on two Cityscapes street scenes. We observe similar behavior as for individual models, where intensity for objects in the distance are sharply peaked, just as the standard deviations computed here. The top image shows how objects close to the ego car have medium variances spread out over a larger area.

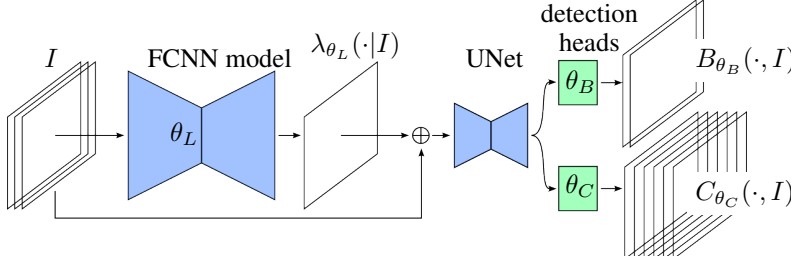

Figure 5: Alternative two-stage object detection architecture based on a first-stage FCNN model for the intensity function. The second stage consists of a UNet encoder-decoder model with two heads, one predicting the spatial extension of bounding boxes and the other one classification. The second stage model is trained separately from first stage on original input images and intensity predictions.

dense intensity function $\lambda_{\theta_L}$ and a second stage $(\theta_B, \theta_C)$ modeling bounding box marks on top of the intensity function.

The training protocol is guided by the interdependency of the parameters $\theta_L, \theta_B$ and $\theta_C$ of the three neural networks $L$, $B$ and $C$. If parameters, e.g., in the model's backbone, are shared, all parameters in $\theta$ can be trained jointly in a one shot manner. However, one can also choose to disentangle the training of $\theta_L, \theta_B, \theta_C$ in stages, if not all parameters are shared. In this work we, follow the latter protocol depicted in Figure 5. The hierarchical model we introduce builds on a semantic segmentation model with one output channel modeling the intensity function $\lambda$ by $\theta_L$. As in the first stage, the regression and classification heads are frozen, the loss function for $\theta_L$ involves only the first line in eq. (7) with object center points as point configuration. We utilize an exponential activation for the intensity function. For the second stage training, we now freeze $\widehat{\theta}_L$. The RGB input image $I$ is concatenated with $\lambda_{\theta_L}$ and fed through a UNet (Ronneberger et al., 2015) architecture (5 stages

Table 6: Calibration values of our PPP method ($\text{ECE}_P$ ↓) and object detector ($\text{ECE}_{BB}$ ↓) for the Cityscapes dataset and different box sizes $s$.

|  | $s$ | 250 | 500 | 750 | 1,000 | 1,500 | 2,500 | 5,000 | 10,000 |
|---|---|---|---|---|---|---|---|---|---|
| DeepLabv3+ | $\text{ECE}_P$ | 0.0012 | 0.0021 | 0.0029 | 0.0030 | 0.0051 | 0.0086 | 0.0120 | 0.0184 |
| HRNet | $\text{ECE}_P$ | 0.0014 | 0.0019 | 0.0027 | 0.0032 | 0.0049 | 0.0085 | 0.0121 | 0.0177 |
| DeepLabv3+ | $\text{ECE}_{BB}$ | 0.0775 | 0.0698 | 0.0625 | 0.0574 | 0.0578 | 0.0567 | 0.0569 | 0.0683 |
| HRNet | $\text{ECE}_{BB}$ | 0.0825 | 0.0750 | 0.0709 | 0.0641 | 0.0647 | 0.0631 | 0.0621 | 0.0750 |

Table 7: Calibration values of our object detector ($\text{ECE}_{BB}$ ↓) trained with $L^2$-loss for the Cityscapes dataset and different box sizes $s$.

|  | 250 | 500 | 750 | 1,000 | 1,500 | 2,500 | 5,000 | 10,000 |
|---|---|---|---|---|---|---|---|---|
| DeepLabv3+ | 0.0795 | 0.0724 | 0.0642 | 0.0587 | 0.0578 | 0.0550 | 0.0541 | 0.0668 |

with a base channel number of 16) for secondary feature extraction after which the features are interpolated back to input shape and concatenated. Two-layer detection heads predict width and height ($\theta_B$) and the class affiliation ($\theta_C$), respectively. The feature maps $B$ and $C$ are evaluated at the point configuration and trained with the second and third line of eq. (7) with available ground truth information. While $B$ and $C$ are computed from the shared parameters in the UNet model, they are disentangled and conditioned on the intensity $\lambda_{\widehat{\theta}_L}$.

The PPP (top) as well as the CMPPP model (bottom) intensity calibration results for the two-stage architecture depending on the test box size are shown in Table 6. For PPP, we observe that calibration for smaller boxes also achieves smaller ECE errors. For CMPPP, we achieve smaller errors, i.e., improved calibrations, compared to semantic segmentation and slightly worse values compared to PPP predictions. While the PPP only predicts the center points of objects, CMPPP adds the prediction of the height and width of the bounding box, which is more challenging for confidence calibration. We observe a similar calibration of the DeepLabv3+ and the HRNet based models. With respect to the bounding box prediction, the DeepLabv3+ based model achieves a mAP of $51.19\%$ and HRNet $54.28\%$.

**Gaussian Residuals.** In Section 3, we discussed a CMPPP model with Laplace-distributed residuals which leads to training based on the $L^1$ regression loss. While this is a widely spread approach to bounding box regression, we may analogously model the residuals as normal distributions by adjusting $p_{\theta_B,\theta_C,\sigma^2}$ in eq. (4). In eq. (12), we then require the cumulative distribution function of the normal distribution with the same mean parameters and variance $\sigma^2$ determined analogously to the Laplace distribution as the mean squared error.

Resulting calibration errors are displayed in Table 7. The calibration values are very similar to those in Table 6. We see small deviations, with training using the $L^1$-loss yielding better values for smaller sampled boxes and slightly worse values for larger ones.

## A.6 VISUAL RESULTS

In Figure 6, we show more qualitative examples, i.e., PPP and CMPPP intensity heatmaps as well as the corresponding bounding box prediction. In Figure 7, visualizations for the VisDrone dataset are given.

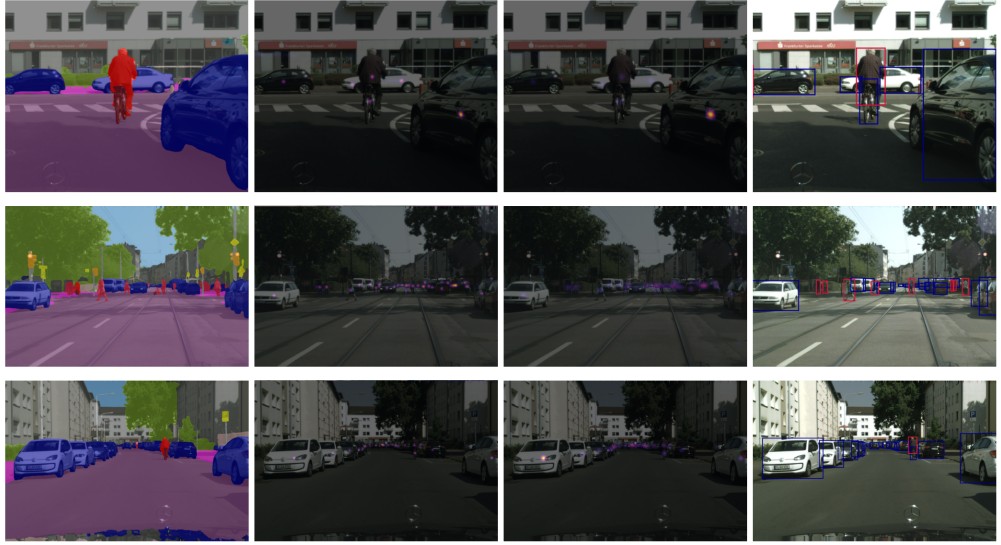

Figure 6: *Left*: Semantic segmentation prediction. *Center left*: Poisson point process intensity. *Center right*: Conditional marked Poisson point process intensity. *Right*: Bounding box prediction.

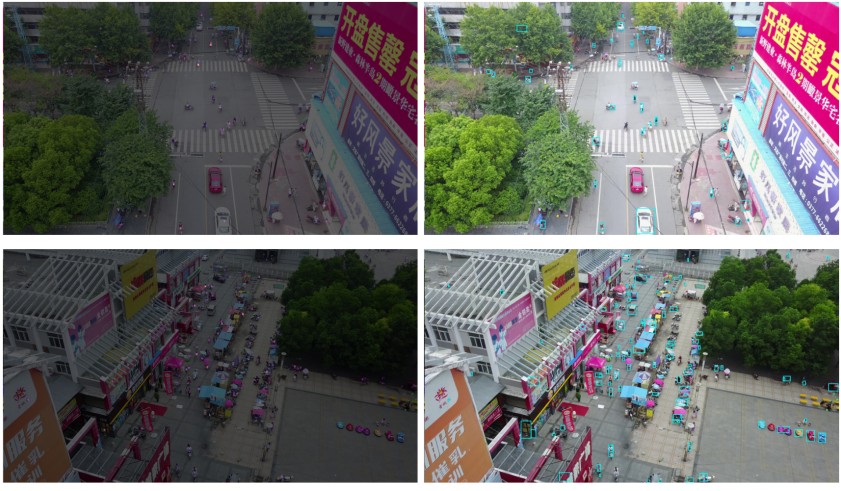

Figure 7: *Left*: Poisson point process intensity. *Right*: Bounding box prediction.

