# OpenReview forum: "Towards Reliable Detection of Empty Space: Conditional Marked Point Processes for Object Detection"
_ICLR.cc/2026/Conference — ICLR 2026 Poster_

### Official Review · Reviewer_dzBf · 2025-10-26

**Soundness:** 3
**Presentation:** 3
**Contribution:** 3
**Rating:** 6
**Confidence:** 2

**Summary:**

This paper proposes a novel object detection framework based on spatial statistics, specifically conditional marked point processes, to address the critical problem of quantifying uncertainty in empty space regions. Unlike traditional object detectors that only provide confidence scores for detected objects, this approach can assess the probability that a given region is truly free of obstacles - a crucial capability for safety-critical applications like autonomous driving. The method models bounding box data as realizations of marked point processes and derives a likelihood-based training objective that enables well-calibrated confidence estimates for empty space regions.

**Strengths:**

1) Novel Problem Formulation and Theoretical Foundation: The paper addresses a fundamental gap in object detection by providing uncertainty quantification for empty space regions, which is crucial for safety-critical applications. The approach is mathematically rigorous, grounded in spatial point process theory, and represents the first application of such methods to deep object detection.
2)Clear Presentation and Methodology: The paper is well-written with clear mathematical exposition, good visualization of results, and comprehensive experimental evaluation across multiple network architectures.

**Weaknesses:**

1) The paper lacks comparison with other uncertainty quantification methods in object detection (Bayesian approaches, ensemble methods, Monte Carlo dropout) and existing calibration techniques. Evaluation is restricted to only two datasets, and there's insufficient analysis of baseline comparisons beyond semantic segmentation models.
2) The Poisson assumption may not hold for real object distributions which often exhibit clustering or repulsion. The factorization in Eq. (3) assumes independence between spatial location and object properties, which may be unrealistic. The method also has scale issues, assigning square patches to detected peaks that conflict with objects of varying sizes.

**Questions:**

1)How realistic is the Poisson assumption for object distributions in real scenes with clustering or mutual exclusion?
2)Have you considered hybrid approaches combining high-performance detectors with your calibration framework?
3)How does computational overhead compare to standard detectors?

---

> ### Author Response · Authors · 2025-11-21
>
> We thank anonymous reviewer dzBf for their valuable feedback and intent to improve our work.
>
> - **W1**: We refer to common response “Re-Calibration” for an extension of our comparison in terms of post-hoc calibration. Further, common response “Datasets” addresses an extension to a third, semantically complementary dataset for indoor robotics. Existing methods like the ones proposed (Bayesian and Bayesian approximations) can be applied to semantic segmentation and objectness baselines but will suffer from the same short-comings as those in terms of void probabilities (see the thought experiment above). While there are models designed to segment drivable areas in images, their effective mechanism is the same as the one in semantic segmentation. In order to reliably assess the safety of certain proposal trajectories in autonomous navigation, it is imperative that an assessment model can be queried about arbitrary image regions. Further, it needs to respond with quantifiable confidence as opposed to binary answers “yes"/"no” such that uncertainty can be faithfully represented. Drivable area predictors can only give such answers on the basis of pixel classification which suffers from the same area scaling problem as semantic segmentation.
> - **W2**: We address the 3 different points made here one by one.
> 	- (also **Q1**) Concerning the realism of our model: Consider an interacting particle model. During inference, finding a likely particle configuration for $n=1,2,3\ldots$ particles has an exponentially fast growing complexity, as for a MAP-predictor you have to query the number of pixels to the $n$-th power of configurations. This is getting intractable soon. Of course, one could then try to apply 'greedy' one particle at a time strategies, but for this there is no strict statistical justification and the real time capability would be problematic if $n$ is high. While generally, the more physically satisfactory model for the intensity function would implement repelling interactions, we overall believe that modeling free particles on camera images is also a good enough and realistic approximation. Our reason for this is that the interactions between particles/bounding boxes are very weak because of the perspective projection onto the optical field of the camera. Due to perspective depth, center points can be close in the image plane while their 2-dimensional bounding boxes overlap quite freely.
> 	- Concerning factorization in equation (3): We do not assume independence here, rather we make the factorization of a joint distribution of $z = (\xi, m)$ to conditional and marginal distribution of $m | \xi$ and $\xi$ a manifest part of our model. There is no independence assumption entering our model.
> 	- The square patch method we use for prediction does not strongly impact the performance of our model as shown in Table 4 in the appendix. By the sharp peaks in the intensity function, large ranges of patch size keep the performance stable.  It is, however, true that the intensity function is taylored to the training resolution to normalize on $n = \mathbb{E}[N]$, the number of ground truth objects. Working with fixed resolutions is, however, not a strong restriction in practice.
> - **Q2**:  Training multi-scale intensity functions in feature pyramids seems like a promising idea which requires, however, careful aggregation of scale-wise intensity functions and re-definition of void confidences. We have refrained from this step in the present work because it is not our primary focus to develop high-performance object detectors if accuracy comes at the ultimate cost of sacrificing the ability to navigate reliably. Our proposed architecture may be regarded minimal in terms of design choice in that only the intensity, class and bounding box feature maps are computed on full input resolution.
> - **Q3**: We discuss the computational overhead of our method and compare with standard object detectors in Section 4.2 “Runtime and scalability".

---

### Official Review · Reviewer_EoWf · 2025-10-31

**Soundness:** 3
**Presentation:** 3
**Contribution:** 2
**Rating:** 6
**Confidence:** 4

**Summary:**

The paper formulates object detection as a conditional marked Poisson point process (CMPPP): box centers are points; widths/heights/classes are marks; an image-conditioned intensity λ and mark distribution p(m|ξ,I) yield a likelihood-based loss and, crucially, closed-form probabilities that arbitrary regions are empty (“drivable”). The loss emerges from the Radon–Nikodym derivative of the (marked) PPP relative to a homogeneous PPP reference, giving a principled alternative to heuristic detection losses. Implementations use segmentation backbones (DeepLabv3+, HRNet, SegFormer) to predict dense maps for intensity, size, and class. Experiments report calibration of void probabilities on Cityscapes and VisDrone and compare mAP to baseline detectors (lower mAP, better void calibration).

**Strengths:**

Principled probabilistic formulation.
- Clear derivation of a likelihood (negative log-RN) for (marked) point processes → a coherent objective for detection and void confidence, instead of ad-hoc objectness + CE/L1. The derivation and discretization details are explicit.

Operational “empty-space” probability.
- Two definitions (no centers in A; no boxes intersecting A) lead to computable expressions, incl. a practical Laplace-based integral for box intersection (Eq. 10–12). This directly targets a safety-critical question standard detectors don’t answer.

Calibration protocol & results.
- Expected Calibration Error (ECE) for randomly sampled boxes across scales; PPP/CMPPP show orders-of-magnitude lower ECE than segmentation-product or standard detectors. Plots and tables are convincing for the posed metric.

Honest positioning.
- Authors explicitly do not claim SOTA mAP and frame the contribution as a probabilistic foundation enabling calibrated emptiness estimates.

For me, this paper look like an "old dish but with a completely new taste." I personally think we need more paper like this.

**Weaknesses:**

Modeling assumptions (independence / PPP).
- A PPP ignores interactions (e.g., repulsion/occlusion between objects). Authors note this in limitations; nonetheless, it undercuts realism and may bias void probabilities in crowded scenes. Extending to Gibbs/repulsive processes or Cox processes would strengthen claims.


Empirical scope is narrow.
- Only two datasets (Cityscapes, VisDrone), limited classes; no distribution-shift tests, no multi-seed variance, and little analysis of sensitivity to discretization (H×W) or the single inference hyperparameter (crop size) beyond an appendix note. Also, more experiments on more standard detection datasets will be helpful, e.g., COCO.

mAP lags standard detectors.
- Reported CMPPP mAP is worse than common baselines like Faster R-CNN/CenterNet; the paper argues calibration is the goal, but many venues expect Pareto curves (mAP vs calibration vs speed) to contextualize trade-offs.

Calibration baseline for segmentation may be weak.
- Their segmentation “void” probability multiplies per-pixel “road” probabilities (independence assumption). Modern segmentation calibration (temperature scaling, Dirichlet, focal-calibration) could shrink the reported gap—this isn’t explored.


Evaluation choices.
- Random box sampling is simple but may not reflect planner-relevant regions (e.g., near obstacles/curbs). Lack of PDQ reporting (they cite PDQ) misses a natural probabilistic detection metric to compare against probabilistic baselines.

Uncertainty taxonomy.
- The method yields aleatoric void probabilities; epistemic uncertainty (e.g., via ensembles/MC-Dropout) is mentioned as combinable but not evaluated. For “reliable” navigation, both matter.

TODOs:
- Trying to extend the model with more advanced architectures, e.g., DINO (the detection one) or latest YOLO.
- Add more experiments on COCO or other larger datasets that covers more diverse objects.

**Questions:**

N/A

---

> ### Author Response · Authors · 2025-11-21
>
> We thank anonymous reviewer EoWf for their valuable feedback and intent to improve our work.
>
> - **W1**: Thank you for taking up this point from our limitations. There are several reasons why we do not model interaction potentials in the present work.
> 	- Introducing Gibbs potentials (a) suffer from the necessity of computing a sum over states bringing the old problems from Boltzmann machines back. Of course, you can avoid evaluation the sum over states by a score based approach, this however would be similar to the likelihood based approach taken in this paper. (b) During inference, finding a likely particle configuration for $n=1,2,3\ldots$ particles has an exponentially fast growing complexity, as for a MAP-predictor you have to query the number of pixels to the $n$-th power of configurations. This is getting intractable soon. Of course, one could then try to apply 'greedy' one particle at a time strategies, but for this there is no strict statistical justification and the real time capability would be problematic if $n$ is high. Taking simple weakly interacting point process models would be feasible, but the correct interaction models would require category specific fine tuning (cars interact differently from humans). We deem this application too domain specific and beyond the scope of this work.
> 	- Moreover, we regard our free particle model as a decent approximation to the interacting particle model. Our reason for this is that the interactions between particles/bounding boxes are very weak because of the perspective projection onto the optical field of the camera. Due to perspective depth, center points can be close in the image plane while their 2-dimensional bounding boxes overlap quite freely.
> 	- Lastly, our focus in this work lied in void probabilities for which the interaction between center points is secondary. Whether a region contains one or more particles is not important to determine whether or not it is drivable.
> - **W2/T2**: We address the different points made here one by one.
> 	- Dataset diversity: We expand our evaluation by experiments on the indoor robotics TiROD dataset, see for this our common response on Datasets. The TiROD dataset contains semantically different categories than Cityscapes and VisDrone which complements our experiments well, covering diverse application domains.
> 	- Distribution shift: This point can be understood in two different ways. A) When it comes to distribution shift to other street scene datasets, we refer to the animation contained in the supplementary material which shows our Cityscapes model evaluated on a KITTI sequence. Our model still reliably detects potential collisions with other traffic participants. B) Domain shift to semantically adverse datasets requires extensive investigations which reach far beyond the scope of our present investigations.
> 	- Multi-seed variance: Reporting multi-seed variance in training of computer vision models is not common practice. During development, we have not noticed significant sensitivity to initialization or data sampling.
> 	- Sensitivity to discretization (H×W): Firstly, we performed the calibration analysis on different test box sizes, see tables in the paper. Secondly, we also reduced the resolution of the Cityscapes images from 1024x2048 to 512x1024 and examined the calibration in the following table (using Deeblabv3+). Even though semantic segmentation is better calibrated due to a flatter softmax distribution caused by the low resolution, we are still significantly better calibrated.
>
> |       | 500    | 750    | 1,000  | 1,500  |
> | ----- | ------ | ------ | ------ | ------ |
> | ECE_S | 0.0553 | 0.0801 | 0.1018 | 0.1344 |
> | ECE_P | 0.0026 | 0.0038 | 0.0057 | 0.0072 |
> - Single inference hyperparameter (crop size) beyond an appendix note: We have included ablation of our only hyperparameter in the appendix (tests across multiple values), as it does not play a role in calibration, but only in the calculation for the mAP, and we obtain stable results across multiple values.
> - More standard detection datasets, e.g. COCO: Evaluating our empty space detection on COCO is not useful, as COCO only annotates object instances and does not contain any semantically defined empty areas; background pixels cannot be interpreted as empty space there. Instead, we use more complex and realistic datasets such as Cityscapes, a drone dataset, and robotic application scenes from TiROD, in which free space is clearly defined and annotated in a way that is meaningful for our method.

---

> > ### Author Response · Authors · 2025-11-21
> >
> > - **W3**: Common object detection baselines tend to finetune objectness thresholds for optimal mAP values. Additionally, since the mAP is quite insensitive to FP predictions, simply choosing excessively small thresholds like $0.1$ or $0.05$ lead to decent mAP measurements. An increase of the number of intensity peaks exhausted to $\mathbb{E}[N] + k$ by our DeepLabv3+ model on Cityscapes val raises the measured mAP to $58.57$ ($k = 10$) which is exactly on par with the baseline models with $59.32\%$ and $57.08\%$ at the expense of introducing a hyperparameter $k$.
> > - **W4**: As explained in our common response above, one cannot expect to obtain satisfactory calibration by applying post-hoc calibration methods to pixel classifiers. For this, we provide experimental evidence. However, we also do observe an improvement in calibration which is certainly worth mentioning for better context of the success of our method.
> > - **W5**: Our visual animation (supplementary material) of intensity function, bounding box prediction and test region void calibration shows a street scene sequence from the KITTI dataset. While trapezoid regions could arguably fit the street in front of the ego vehicle more tightly, our rectangular region covers a significant part of the relevant street and responds well to slight overlaps with predicted bounding boxes on the side of the road.
> > - **W6**: We believe that the extension to a sampling-based ensemble model will not significantly strengthen our method presently as we have shown vast improvements over existing methods. Since ensemble variance over predictions are hardly interpretable without tuning a variance threshold, epistemic uncertainty measures offer little benefit to our goal. The only benefit that can currently be expected from sampling is to have improved intensity robustness. In our current investigations, this is, however not an issue. Empty-space prediction is dominated by scene geometry, occlusions, and sensor noise—factors that are intrinsically stochastic but not reducible through additional model knowledge. Epistemic uncertainty (e.g., via ensembles or MC-Dropout) is therefore not informative for this task: reducing parameter uncertainty does not change the fundamental ambiguity of whether a region is physically free. For this reason, we focus on aleatoric uncertainty, which directly corresponds to the safety-relevant ambiguity in empty-space estimation.
> > - **T1**: An extension of our model to the object detectors mentioned could potentially lead to strong state-of-the-art models capable of both, detecting foreground objects with high accuracy and making well-calibrated void predictions. It is, however, not our primary focus to develop high-performance object detectors if accuracy comes at the ultimate cost of sacrificing the ability to navigate reliably. Our proposed architecture may be regarded minimal in terms of design choice in that only the intensity, class and bounding box feature maps are computed on full input resolution. Adapting existing object detection models to our method involves the computation of additional full-resolution feature maps to model an intensity function and is, therefore, conceptually akin to the design of panoptic segmentation.

---

> ### Author Response · Authors · 2025-12-02
> **Addressing W2 (multi-seed variance) and W6**
>
> We additionally evaluated a small ensemble of independently trained CMPPP models (DeepLabv3+) on the Cityscapes dataset. We performed two complementary evaluations: (1) we computed mAP and ECE for each ensemble member and averaged the resulting metrics, and (2) we first averaged the ensemble's predictions (intensity, width, height and softmax feature maps) and then computed the metrics on the aggregated prediction.
>
> An ensemble of $5$ CMPPP models (1) achieves an averaged mAP of $51.46 \pm 0.82 \%$, while the ensemble CMPPP model (2) yields a significantly improved mAP of $54.29\%$. Meanwhile the bounding box void ECE akin to table 3 with $s = 1.\ 000$ stays at $0.10$ (standard deviation across ensemble members remains very small at about $1\%$), showing improved performance under preserved calibration properties.
>
> Evaluation (1) also addresses the question of sensitivity of the model with respect to random seeding of the model training and we demonstrate that both, object detection performance and calibration, remain robust under different seeds.

---

### Official Review · Reviewer_hbUL · 2025-11-01

**Soundness:** 2
**Presentation:** 3
**Contribution:** 2
**Rating:** 2
**Confidence:** 4

**Summary:**

The paper proposes a Conditional Marked Poisson Point Process (CMPPP) model for object detection. The core motivation is to provide a probabilistically sound framework that can accurately estimate the confidence of "empty space" (drivable areas), addressing the lack of such uncertainty measures in standard object detectors. The method is derived from spatial statistics, using a negative log-likelihood loss for end-to-end training. Experiments on Cityscapes and VisDrone compare the proposed method's calibration against semantic segmentation and older object detection baselines.

**Strengths:**

- Theoretical Novelty: The derivation of the object detection task from the theory of marked point processes is mathematically grounded and offers a different perspective compared to standard heuristic-based loss functions.

- Addressing an Overlooked Problem: attempting to quantify the uncertainty of regions without detections is a relevant topic for safety-critical applications.

**Weaknesses:**

- Questionable Problem Formulation: The paper heavily prioritizes calibration over standard accuracy metrics. However, calibration does not mean high accuracy. The premise that drivable area requires such a complex probabilistic setup is not entirely convincing; in many standard applications, drivable area is effectively treated as a discrete distribution for decision-making. The experimental setting for calibration appears somewhat contrived to highlight the proposed method's strengths while ignoring standard operational requirements (high mAP).

- Missing Standard Calibration Techniques in Baselines: The paper compares its intrinsically calibrated method against standard DNNs (like DeepLabv3+) that are known to be miscalibrated out-of-the-box. A fair comparison requires these standard models to be evaluated with common post-hoc calibration techniques applied, most notably temperature scaling. It is possible that a standard detector with simple temperature scaling achieves comparable empty-space calibration to the proposed complex CMPPP method, which would significantly diminish the core contribution.

- Unfair Baselines (Segmentation Task): The comparison with semantic segmentation models regarding "drivable area" calibration is unfair. The baselines were trained in a multi-class setting. For a fair comparison, the semantic segmentation baselines should be trained specifically in a binary classification setting (road vs. non-road).

- Outdated Baselines (Detection Architectures): The chosen object detection baselines (Faster R-CNN, CenterNet) are outdated for ICLR 2026. The field has moved to transformer-based architectures. The authors should compare against DETR or its more recent variants to demonstrate if the proposed CMPPP really offers advantages over modern state-of-the-art detectors.

**Questions:**

- Why did you not include temperature scaling (or other standard post-hoc calibration methods) for the baseline models? Comparing against uncalibrated raw logits is a weak baseline.

- Why did you not train the semantic segmentation baselines on the binary "road vs. non-road" task for a fairer comparison of empty space calibration?

- Can you provide results comparing your method to modern detection architectures like DETR?

- Given that calibration is not a substitute for accuracy, how does the downstream planner benefit from a well-calibrated but less accurate detector?

---

> ### Author Response · Authors · 2025-11-21
>
> We thank anonymous reviewer hbUL for their valuable feedback and intent to improve our work.
>
> - **W1**: We argue that the focus of conventional object detectors on high accuracy and good objectness calibration misses the point of what confidence is actually meaningful for reliable trajectory planning. While high mAP is a good proxy for object detection performance agnostic with respect to application domain, autonomous agents need to calibrate the statement “Trajectory (i.e., input region) $A$ is free from collisions.” Object detectors can only answer this in a binary fashion with “yes” or “no” by checking intersection with predicted objects, but give no non-binary confidences (e.g., “$A$ is collision-free with probability $\widehat{\mathbb{P}}(N(A) = 0) = 0.00073$.”). What is calibrated in object detection is the statement “The image region corresponding to anchor/centerness region $B$ contains an object.” which focuses on existence of foreground rather than $B$ being void. One can now derive the complementary statement about a test region $A$ being void from, say, the spatial objectness heatmap of a region proposal network. The logically (according to the probabilistic formulation of the model) aggregated void confidences, say, from anchors $B$ that intersect with $A$, has been shown to be highly mis-calibrated in our experiments for standard architectures.
> - **W2/Q1**: We refer for this point to our common response above on re-calibration. Although calibration is slightly improved, the obtained confidences remain at least an order of magnitude worse calibrated than our model. Note, that we observe that the semantic segmentation models at pixel level (s=1) are comparatively well calibrated in comparison to larger test boxes.
> - **W3/Q2**: For a binary segmentation model, the same reasoning as in our previous explanation in the common response on re-calibration. Even for a semantic segmentation model only trained on 2 classes, we cannot expect sensible void calibration from pixel classifiers due to the issue of test area scaling. Note that our semantic segmentation model DeepLabv3+ achieves an IoU of $98.02\%$ on the road class on Cityscapes val which indicates high accuracy for classifying road pixels (accuracy of $99.1\%$). We believe that a model that trains only on 2 classes will not differ significantly.
> - **W4/Q3**: We are aware of modern transformer models for object detection and their superior accuracy. The short answer why we do not consider them here (also for comparisons) is, that they do not model a relation of spatial location and objectness in any transparent way due to their design and one cannot derive void confidences from them in any natural way. On a more elaborate note: The main functionality of DeTR is to query a latent token representation with $N$ positional encodings and subsequently perform classification with a “background” option and regressing bounding box features. While on token level, the background confidence relates to the absence of objects, it is ambiguous how to aggregate them over a fixed test region $A$ of the image. In contrast, feature maps like RPN objectness in convolutional neural networks at least preserve the spatial relation with the input, such that void confidences can be derived depending on $A$. That is, why we use them for comparison even though we are aware that they still suffer from the scaling problem of the test area.
> - **Q4**: Any object detector produces prediction errors except on unrealistic toy data. We, therefore argue that the prediction of a highly accurate object detector is not beneficial for trajectory planning due to the absence of quantifiable collision assessment. Object detectors can only ever report on the danger of possible collision in terms of “yes” and “no” but not in a quantified or reliable way that can reasonably be tested in a calibration experiment. From this point of view, our contribution constitutes a usable and more interpretable module in the downstream pipeline. Precise bounding boxes are not useful for trajectory planning if they do not allow for reliable and quantifiable collision avoidance; both of which our model provides.

---

### Official Review · Reviewer_PjVJ · 2025-11-02

**Soundness:** 3
**Presentation:** 3
**Contribution:** 2
**Rating:** 4
**Confidence:** 4

**Summary:**

In this paper, the authors tackles the problem that modern object detectors provide confidence only for detected objects but not for empty regions. The authors propose a probabilistic approach based on Conditional Marked Poisson Point Processes (CMPPP) that are able to model both detections and confidently predict empty spaces. By treating object centers as spatial points with marks for size and classes, the model can estimate the probability that any region is truly object-free. Trained with corresponding likelihood loss, the authors demonstrate that their approach results in well-calibrated aleatoric uncertainty and achieves competitive detection accuracy.

**Strengths:**

* The paper contributes to the important study of uncertainty-based object detection, which is highly relevant for autonomous driving and robotics applications.

* It is clearly written and easy to follow; the main idea of the proposed method is intuitive, and the limitations of prior approaches are well described.

* The proposed probabilistic framework is principled and mathematically grounded, providing a coherent way to quantify uncertainty for both detected objects and empty regions.

**Weaknesses:**

* The experimental evaluation is relatively narrow, focusing mainly on the Cityscapes dataset; testing on additional datasets (e.g., KITTI, BDD100K) would better demonstrate generalization to diverse environments and scene layouts. The same experimental protocol could also be extended to 3D object detection tasks using datasets such as nuScenes or Waymo, which would show whether the proposed probabilistic modeling scales to spatially richer domains.

* The claimed improvement in calibration would be more convincing with comparisons to strong post-hoc calibration baselines (e.g., temperature scaling) applied to existing detectors with center-prediction segmentation heads (CenterNet-style), where re-calibration could be performed pixel-wise. While such methods may require a separate calibration set, this limitation can often be mitigated in practice. To strengthen the claim, the paper could include an analysis of how much calibration data would actually be needed to achieve comparable performance with standard post-hoc approaches.

* While the method enables probabilistic estimation of empty-space confidence, its practical relevance remains unclear. The paper does not demonstrate how this “emptiness calibration” translates to downstream tasks such as planning, risk estimation, or control. To make the contribution more impactful, the authors could connect the calibrated emptiness probabilities to decision-making metrics — for instance, by integrating them into a planner or trajectory evaluation module. Extending the framework to 3D object detection and testing on datasets like nuScenes or Waymo would also allow assessing how such uncertainty information might affects predicted vehicle trajectories and safety-related metrics.

**Questions:**

* Have you evaluated how well the method generalizes beyond Cityscapes, for instance on KITTI or BDD100K, or considered extending it to 3D datasets like nuScenes or Waymo?

* How would your approach compare to standard post-hoc calibration methods such as temperature scaling or pixel-wise calibration applied to CenterNet-style detectors?

* How could the proposed emptiness calibration be integrated into downstream tasks like motion planning or trajectory evaluation to demonstrate its practical value?

---

> ### Author Response · Authors · 2025-11-21
>
> We thank anonymous reviewer PjVJ for their valuable feedback and intent to improve our work.
>
> - **W1/Q1**: Due to the semantic and scene similarity between Cityscapes, KITTI and BDD we decided to extend our investigations rather to indoor scenes for robotics applications from the TiROD dataset. See the common response above for details on our extended evaluation. Regarding 3D object detection: A meaningful model of an object intensity function in 3D is a natural further step but requires highly nontrivial adaptations and reformulations of the method to either lidar or radar signals which are beyond the scope of our paper.
> - **W2/Q2**: We refer for this point to our common response above on re-calibration. The problem with calibration for CenterNet comes from the fact that uncertainty relates to a question and a prediction. As CenterNet offers no prediction for the probability that a region is not occupied, you also cannot scale/re-calibrate this. Taking the centerness scores as a surrogate for the pixel-wise prediction, you end up in the same problem we found for semantic segmentation. On the pixel level, these models *are* calibrated. But the more pixels a region contains, the less calibrated are pixel-based predictions. Consider a thought experiment and let the resolution of the image tend to infinity. In this limit, the only calibrated pixel-based models would be the perfect models that never err. All other perception networks that are calibrated on the pixel level will predict zero probability for empty space for any region. See the table below for the numerical manifestation of this effect. To elaborate this central aspect further, we investigated the effect of post-hoc calibration methods experimentally.
> - **W3/Q3**: Regarding down-stream application: The application to trajectory planning based on camera images is evident by estimating ego motion and the extrapolated space that the ego vehicle will occupy in the current field of vision. The probability of (an occupied region of) a trajectory remaining collision free within the current scene is given by eq. 10 and 11 and has been shown to be well-calibrated in our experiments. Our calibration results also include those cases where our CMPPP object detector has overlooked obstacles which lead to collisions. Still, our model yields well-calibrated confidences.

---

### Author Response · Authors · 2025-11-21
**Common Reponse on Comparison with Re-Calibration Methods**

We thank all reviewers for their valuable feedback. Two points came up more than once, so we decided to address them as common responses to all reviewers.

We concede that a better description of the problem and corresponding experimental validation is in order:

Our motivation for applying spatial point process models is that void confidences from pixel classifiers such as semantic segmentation or objectness cannot be rectified by re-calibration techniques. For empirical validation of this claim, we provide results for semantic segmentation and objectness post-hoc calibration which show that satisfactory re-calibration of the obtained confidences is not possible, even when the area of test regions is fixed. Below, we provide an intuitive conceptual explanation of this phenomenon.

In the following table, we tested temperature scaling (ts) calibrations. Although this slightly improved void calibration of the baseline models, they remain at least an order of magnitude worse calibrated than our model.

| Model      | Metric   | s=1    | 250    | 500    | 750    | 1000   | 1500   | 2500   | 5000   | 10000  |
| ---------- | -------- | ------ | ------ | ------ | ------ | ------ | ------ | ------ | ------ | ------ |
| DeepLabv3+ | ECE_S    | 0.0591 | 0.1102 | 0.1741 | 0.2033 | 0.2245 | 0.2521 | 0.2667 | 0.2417 | 0.1948 |
|            | ECE_S_ts | 0.0514 | 0.1286 | 0.1411 | 0.1525 | 0.1618 | 0.1653 | 0.1797 | 0.1837 | 0.0922 |
|            | ECE_P (ours)    | 0.0000 | 0.0012 | 0.0017 | 0.0029 | 0.0029 | 0.0046 | 0.0062 | 0.0109 | 0.0164 |
| HRNet      | ECE_S    | 0.0682 | 0.0413 | 0.0859 | 0.1142 | 0.1443 | 0.1785 | 0.2206 | 0.2295 | 0.1939 |
|            | ECE_S_ts | 0.0609 | 0.0378 | 0.0596 | 0.1245 | 0.1329 | 0.1397 | 0.1520 | 0.1597 | 0.1641 |
|            | ECE_P (ours)    | 0.0000 | 0.0008 | 0.0012 | 0.0014 | 0.0019 | 0.0022 | 0.0041 | 0.0053 | 0.0071 |
| SegFormer  | ECE_S    | 0.0755 | 0.0621 | 0.0585 | 0.0609 | 0.0593 | 0.0588 | 0.0582 | 0.0626 | 0.0713 |
|            | ECE_S_ts | 0.0863 | 0.0434 | 0.0378 | 0.0379 | 0.0338 | 0.0344 | 0.0333 | 0.0364 | 0.0481 |
|            | ECE_P (ours)    | 0.0000 | 0.0006 | 0.0008 | 0.0014 | 0.0018 | 0.0027 | 0.0046 | 0.0053 | 0.0082 |

**Semseg re-calibration**: We fit a temperature scaling to $20\%$ of the Cityscapes validation data and evaluate on the complementary set. For each value of $s$, we fit and evaluate on test boxes of that size exclusively as this is a best case scenario for the re-calibration of pixel classifiers. No generalization to other (significantly larger or smaller) test box sizes is required. Realistically, calibration would be performed at the pixel level and then tested on other sizes, which yields poorer results than scaling for the respective test box size as shown in the table.

**OD re-calibration**: We perform the same re-calibration scheme for our object detection baselines (Faster R-CNN and CenterNet). The ECE value for both models tested for a sample box size of $s=1000$ improves from $0.9915$ when uncalibrated to $0.5435$ with temperature scaling. We observe that our method is significantly better calibrated than segmentation or object detection models.


**Explanation**: The fact that this is the case can be understood by the following thought experiment. For test region $A$, a semseg or objectness model predicts some void confidence $\widehat{p}(A)$ as a product of confidences/objectness values over (super-)pixels within $A$ as described in our paper. We show that, and argue why, these values are vast under-estimations of the measured void frequencies. Increasing the area by a factor of 2 to an area $A'$ will also increase the number of factors in the confidence multiplication by 2, so we expect a behavior of $\widehat{p}(A') \approx [\widehat{p}(A)]^2$, i.e., we have exponential scaling with respect to the increasing factor of the area.
One the one hand, this means that it is impossible to re-calibrate well across significantly differing test region sizes. On the other hand, even for fixed areas of test regions $A$, the confidences obtained from a pixel classifier by multiplying per-pixel confidences cannot even be re-calibrated to a satisfactory degree. This is because the model is not designed to yield meaningful confidences across image neighborhoods.

We believe that making this argument more explicit and plastic in our introductory section strengthens the motivation for our model design and experimental evaluation. We will include the experimental validation of this intuition in section 4.2.

---

> ### Author Response · Authors · 2025-11-21
> **Common Reponse on Dataset Extension**
>
> As our evaluation already encompasses one dataset from the autonomous driving area (Cityscapes) and one from drone navigation (VisDrone), we decide to extend our results by another dataset of indoor scenes targeted to robotics applications, the TiROD dataset (https://arxiv.org/html/2409.16215v1). The dataset contains 13 semantically different classes ('bag', 'bottle', 'cardboard box', 'chair', 'potted plant', 'traffic cone', 'trashcan', 'ball', 'broom', 'garden hose', 'bucket', 'bicycle', 'gardening tool') than our current datasets Cityscapes and Visdrone. Natively split into 5336/662/666 train/test/val samples, the TiROD dataset serves as a further use case scenario for autonomous navigation. The following table shows calibration values for Deeplabv3+ across different sizes of the test box. We are also observing minor ECE errors for this dataset.
>
> |        | 500    | 750    | 1,000  | 1,500  |
> | ------ | ------ | ------ | ------ | ------ |
> | ECE_P  | 0.0043 | 0.0056 | 0.0083 | 0.0115 |
> | ECE_BB | 0.0954 | 0.1019 | 0.1092 | 0.1237 |

---

### Meta-Review · Area_Chair_FFH2 · 2026-01-02

**Summary:**

The core contribution of the paper is a principled reformulation of drivable space (“empty-space”) estimation as a probabilistic spatial event, enabling calibrated void probabilities for arbitrary regions. These capabilities are difficult to obtain reliably from pixel-wise aggregation or standard detection confidence scores.

At the same time, the empirical validation remains limited. In particular, the evaluation scope is narrow in terms of dataset diversity and modern detection baselines, the reported mAP lags behind standard detectors without a clear Pareto analysis of accuracy-calibration trade-offs, and the downstream relevance to planning is only indirectly demonstrated. Moreover, the reliance on a non-interacting Poisson point process (i.e., without explicit modeling of object–object interactions) may limit realism in crowded scenes, and the calibration baselines for segmentation and detection could be strengthened with more competitive post-hoc or probabilistic alternatives.

The initial reviewer scores were 6, 6, 4, and 2 (average: 4.5), slightly below the acceptance threshold. While the rebuttal does not fully resolve all concerns regarding experimental breadth and downstream validation, the paper presents a novel and mathematically grounded problem formulation with a clear, principled probabilistic framework that targets a safety-relevant capability (calibrated emptiness for arbitrary regions) that standard detectors are not designed to provide. Given the potential value of this perspective to the community despite remaining empirical limitations, I recommend Accept.

**Reviewer Concerns:**

The authors provide clear rebuttals and reasonable justifications for their design choices. In particular, the addition of post-hoc calibration experiments (temperature scaling) helps substantiate the limitation of pixel-level re-calibration for void confidence estimation, and the inclusion of experiments on the TiROD dataset positively demonstrates that the proposed calibration behavior generalizes beyond the originally evaluated domains.

However, despite showing some performance improvements through ensemble experiments, several reviewer requests remain insufficiently addressed empirically. In particular, the evaluation was not extended to widely used driving datasets such as KITTI or BDD, comparisons against modern transformer-based detection architectures, and quantitative validation on downstream tasks such as planning are largely discussed at a conceptual level. The absence of such practical and task-level evaluations limits the assessment of the proposed model’s applicability to real-world autonomous driving systems.

**Reviewer Scores:**

The initial reviewer scores were 6, 6, 4, and 2 (average: 4.5), slightly below the acceptance threshold. No additional comments were provided by the reviewers after the authors’ rebuttal. While the proposed method is not exhaustively validated experimentally, the paper presents a strong novel problem formulation and a solid theoretical foundation.

---

### Decision · Program_Chairs · 2026-01-26

Accept (Poster)